# Rare-mark-aware Next Event Prediction in Marked Event Streams

## Abstract

In marked event streams, Marked Temporal Point Process (MTPP) is central to predicting when and what mark the next event will occur based on the history. In various real-world applications, the mark distribution is significantly imbalanced, i.e., some marks are frequent, and others are rare. We unveil that such imbalance can cause the rare mark missing issue when predicting the next event – frequent marks are dominant, and rare marks often have no chance. However, rare marks can be essential in some applications (e.g., the occurrence of a 7-magnitude earthquake), and missing such rare marks in the next event prediction is risky. To address this issue, we tackle a novel Rare-mark-aware Next Event Prediction problem (RM-NEP), answering two questions for each mark $m$: *"what is the probability that the mark of the next event is $m$?"* and *if $m$, when will the next event happen?"*. Solving RM-NEP gives rare marks equal opportunity as frequent marks in the next event prediction. This guarantees that rare marks are always included in the predicted results. Moreover, RM-NEP allows arbitrary number of rare marks samples for time prediction without interference from frequent marks, ensuring the time prediction is accurate. To solve RM-NEP effectively, we first unify the improper integration of two different functions into one and then develop a novel Integral-free Neural Marked Temporal Point Process (IFNMTPP) to approximate the target integral directly. Extensive experiments on real-world and synthetic datasets demonstrate the superior performance of our solution for RM-NEP against various baselines.

## 1 Introduction

Events have been generated continuously in human activities or observed from natural phenomena. The Temporal Point Process (TPP) models point sequences that represent the arrival time of events. TPPs are built upon rich theoretical foundations, with early work dating back to many decades ago, where they were used to model the arrival of insurance claims and telephone traffic(Shchur et al., 2020), till now widely applied in social network analysis(Farajtabar et al., 2017; Rizoiu et al., 2017; Zeng & Gao, 2022), neural logic inference(Mei et al., 2020; Li et al., 2020; 2022), and biological activity modeling(Tagliazucchi et al., 2012).

The Marked Temporal Point Process (MTPP) models scenarios where each event comes with a mark and its arrival time. The mark can be categorical, such as the small/large earthquakes, mild/moderate/critical and symptoms of patients visiting an emergency department, sell/buy in financial transactions; it can also be numerical, such as temperature in the weather forecast, the longitude and latitude of observations in ecology. As often encountered in practice, the MTPP has attracted much attention from the research community (See Shchur et al. (2021) for a comprehensive review). Most existing studies assume that events in sequences are correlated, and therefore MTPPs are conditioned on history, i.e., the events that occurred so far.

MTPP is central to modeling the sequence of events and predicting the mark and time of the next event, based on the conditional joint Probability Distribution Function (PDF), denoted as $p^*(m, t)$ [1], learned from history by encoding the interconnection between event mark $m$ and inter-event time $t$. Typically, a single mark and a single time are returned as the prediction. For the sake of description,

---

[1]The asterisk reminds the probability is conditioned on history.

we name it Next-event Prediction problem (NEP) in this study. Most existing MTPP studies first predict when the next event will occur and then what mark the next event is at the time predicted(Mei & Eisner, 2017; Zhang et al., 2020; Zuo et al., 2020; Mei et al., 2022). Few studies first predict what the mark of the next event is and then when it will occur(Waghmare et al., 2022). Some recent studies solve NEP by directly generating the time and mark of the next event simultaneously(Yuan et al., 2023; Lüdke et al., 2023).

In various real-world scenarios, the mark distribution is significantly imbalanced, i.e., some marks are highly frequent and others are rare. The distribution imbalance may significantly impact the solution of NEP - the frequent marks are dominant so that rare marks have no chance in the next event prediction. However, rare marks are often significant for real-world applications, and neglect of such rare marks is risky. Let us consider the following scenario. Major earthquakes with significant magnitude (for example, bigger than 7.0 on the Richter scale) are rare but considered devastating as they can cause heavy damage to the city and lots of casualties[2]. Therefore, with an MTPP-based earthquake predictor for a region, people are interested in when it will occur if the next earthquake is a major one, even rare but possible. However, the small earthquakes are frequent. Following the NEP, it predicts when the next earthquake will happen, denoted as $t$, and then what kind of earthquake it might be at $t$. As shown in Figure 1, small earthquakes will dominate the next event prediction and the major ones may be never predicted, making it impossible to inform the time of major earthquake if it will occur next. We name it the *rare mark missing* issue in NEP.

Our study shows that the rare mark missing issue is intrinsic in NEP and is hard to tackle if predicting a single mark and a single time like NEP. To address the root cause of the rare mark missing issue, we propose the new Rare-mark-aware Next Event Prediction problem (RM-NEP). Different from NEP, RM-NEP gives rare marks an equal chance as frequent marks in the next prediction by answering two questions for each mark $m$ "*what is the probability that the mark of the next event is $m$? and "if $m$, when will it happen?* This guarantees that rare marks are consistently included in the predicted results. Furthermore, RM-NEP allows an arbitrary number of samples for rare marks in the next event time prediction without the interference of frequent marks, so the time prediction for rare marks is accurate. Solving RM-NEP faces the unique challenge of improper integration over the infinite time interval for estimation of the probability of marks and their time. Classical numeric integration methods such as Monte Carlo integration are computationally heavy and can only estimate integrals on a finite interval. To attack the challenge, we first propose to unify two improper integral functions into one and then develop a novel MTPP model specifically designed to efficiently solve improper integral function, called IFNMTPP (Integral-free Neural Marked Temporal Point Process). Contributions of this study are threefold:

- This study identifies the rare mark missing issue in the Next-event Prediction problem (NEP) and propose a novel Rare-mark-aware Next Event Prediction problem (RM-NEP) to give rare marks equal chance as frequent marks in the next event prediction.

- For an efficient solution of RM-NEP, we unify the two improper integral functions involved into one.

- To improve the improper integration, we develop a novel model IFNMTPP to directly approximate the integration via a simple monotonically decreasing neural network.

## 2 PRELIMINARIES AND PROBLEM STATEMENT

### 2.1 PRELIMINARIES

The Marked Temporal Point Process (MTPP) is a random process whose embodiment is a sequence of discrete events, $\mathcal{S} = \{(m_i, t_i)\}_{i=1}^l$, where $i \in \mathbb{Z}^+$ is the sequence order, $t_i \in \mathbb{R}^+$ is the time when the $i$th event occurs, $m_i$ is the mark of the $i$th event. This study only concerns a finite set of categorical marks $\mathrm{M} = \{k_1, k_2, \cdots, k_{|\mathrm{M}|}\}$, and the simple MTPP, which allows at most one event at every time, thus $t_i < t_j$ if $i < j$. The time of the most recent event is $t_l$, and the current time is $t > t_l$. The time interval between two adjacent events is the inter-event time. We assume that an event with a particular mark at a particular time may be triggered by past events. Let $\mathcal{H}_{t_l}$ be the history up to (including) the most recent event, and $\mathcal{H}_{t-}$ be the history up to (excluding) the current

---

[2]http://earthquake.usgs.gov/earthquakes/eqarchives/year/eqstats.php

time(Rasmussen, 2018). With these definitions, we can define the Conditional Intensity Function (CIF) of MTPP:

$$\lambda^*(m = k_i, t) = \lambda(m = k_i, t|\mathcal{H}_{t-}) = \lim_{\Delta t \to 0} \frac{P(m = k_i, t \in [t, t + \Delta t]|\mathcal{H}_{t-})}{\Delta t}. \tag{1}$$

With $\lambda^*(m, t)$, the conditional joint PDF of the next event can be defined:

$$p^*(m, t) = p(m, t|\mathcal{H}_{t_l}) = \lambda^*(m, t)F^*(t) = \lambda^*(m, t)\exp(-\int_{t_l}^t \sum_{n \in \mathrm{M}} \lambda^*(n, \tau)d\tau). \tag{2}$$

where $\tau$ means time. $F^*(t)$ is the conditional PDF that no event has ever happened up to time $t$ since $t_l$. The detailed elaboration of how to obtain Equation (2) from Equation (1) is in Appendix A.

The simplest form of MTPP is the homogeneous Poisson process whose CIF merely contains a positive number, i.e., $\lambda^*(m = k_i, t) = c$. Another example is the Hawkes process(HAWKES, 1971), belonging to the self-exciting point process family. Its CIF is $\lambda^*(m = k_i, t) = \mu_i + \sum_{j:t_j < t} \kappa_i(t, t_j)$ where $\kappa_i(t, t_j) > 0$ represents the excite from previous events. Because it meets the real-world intuition that the influences of occurred events always drastically drops as time passes, the Hawkes process is a widely used backbone process in various models(Cao et al., 2017; Salehi et al., 2019; Arastuie et al., 2020; Li & Ke, 2020; Okawa et al., 2021; Idé et al., 2021; Huang et al., 2022).

With MTPP, NEP requires a single mark and a single time as the prediction of the next event. The first method for solving NEP, utilized by most existing MTPP approaches, predicts when the next event will occur and then its mark. Conceptually, the expected time of the next event is $\bar{t} = \int_{t=t_l}^\infty \tau p^*(\tau)d\tau$ where $p^*(t) = \sum_{m \in M} p^*(m, t)$. The numerical method is typically used to calculate $\bar{t}$ by sampling $N$ times, denoted as $\{t^i\}_N$, from distribution $p^*(t)$ following Thinning Algorithm (TA) or Inverse Transform Sampling (ITS)(Rasmussen, 2018) so that $\bar{t} = \frac{1}{N} \sum_i t^i$. Then, mark of the next event at $\bar{t}$ is predicted by $m_{\bar{t}} = \arg\max_{m \in M} p^*(m, \bar{t})$. The second method first predicts the mark of the next event $m = \arg\max_{m \in M} p^*(m)$ and then predicts time of the next event $\bar{t}_m = \int_{t=t_l}^\infty \tau p^*(\tau|m)d\tau$(Waghmare et al., 2022). The third method is to generate the time and mark of the next event simultaneously. Yuan et al. (2023) and Lüdke et al. (2023) implement this method by training a DDPM to enable sampling $p^*(m, t)$ from a normal distribution.

## 2.2 NEP AND RARE MARK MISSING

With MTPP, NEP requires a single mark and a single time as the prediction of the next event. The frequent marks dominate the results. Let us consider two marks $k_1$ and $k_2$ where $k_1$ is much more frequent than $k_2$ in the observed event sequence. Suppose the next event is mark $k_2$. Because $k_1$ is much more frequent than $k_2$, it is very likely that $p^*(k_1, t) > p^*(k_2, t)$ for most time $t$, including $\bar{t} = \int_{t=t_l}^\infty \tau \sum_{m \in M} p^*(m, \tau)d\tau$. If so, $k_1$ will be predicted as the next event. In the extreme case, if $p^*(k_1, t) > p^*(k_2, t)$ for every time $t$, $k_1$ will always be predicted as the mark of the next event, while $k_2$ will never have the chance. This is true no matter the order that the time and mark of the next event are predicted. We name this situation as *rare mark missing* in the results of NEP.

Figure 1 (a) demonstrates the mark frequency distribution in four datasets (see details in Section 4). Figure 1 (b) shows $p^*(m, t)$ for each mark $m$ in these datasets. The envelope covers $p^*(m, t)$ of all instances in the datasets and the line is the average of $p^*(m, t)$ across these instances. Figure 1 (c) presents the percentage of each mark in the results of NEP. Using *Retweet* as an example, the frequent marks (mark 0 and 1 occupying around 50% and 45% of data) have a much higher $p^*(m, t)$ than the rare mark (mark 2 occupying around 5%) at almost every time $t$ for almost all instances as shown in Figure 1 (b) (leftmost). So, marks 0 and 1 are predicted as the next event mark while mark 2 has no chance, as evidenced in Figure 1 (c) (leftmost). A similar situation can be observed on other datasets with other MTPP approaches.

## 2.3 PROBLEM STATEMENT

The Rare-mark-aware Next Event Prediction problem (RM-NEP) aims to allow arbitrary number of rare marks samples for time prediction without the interference of frequent marks. Solving RM-NEP

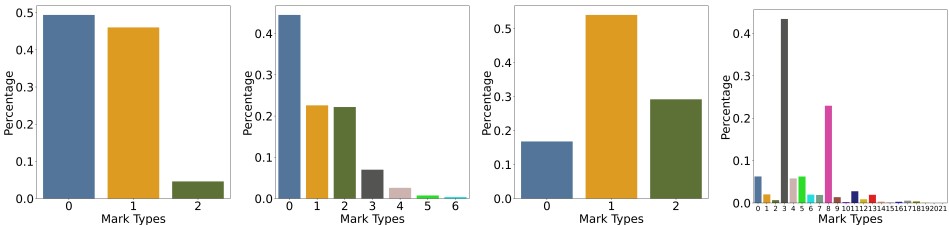

(a) The frequency distribution of marks in Retweet, USearthquake, Yelp, and StackOverflow(from left to right).

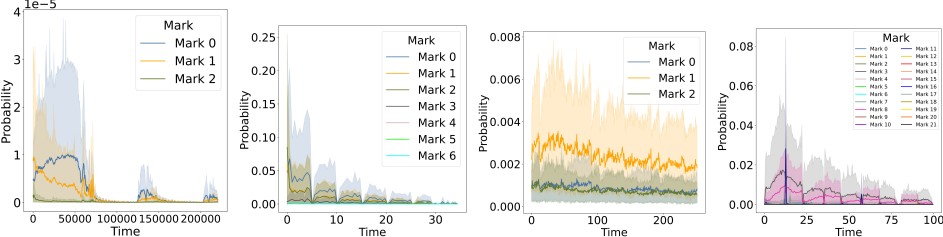

(b) The $p^*(m, t)$ for each mark $m$ in Retweet, USearthquake, Yelp, and StackOverflow(from left to right).

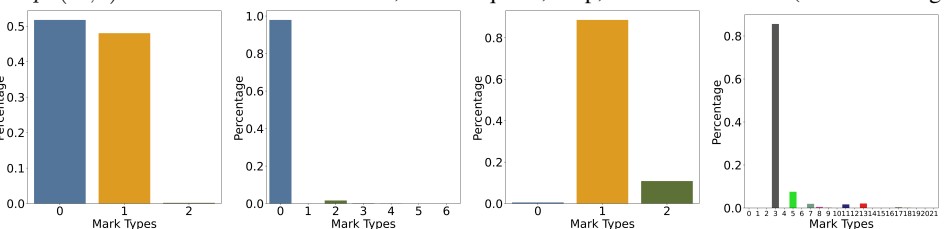

(c) The frequency distribution of marks in the results of NEP (i.e., predicted as the next event mark) using SAHP(Zhang et al., 2020) in Retweet, USearthquake, Yelp, and StackOverflow(from left to right).

Figure 1: A demonstration of rare mark missing in the results of NEP on four real-world datasets. A similar situation can be observed with other MTPP approaches.

answers two questions for each mark $m$: "*what is the probability that the mark of the next event is $m$?*" and "*if $m$, when will the next event happen?*". The RM-NEP can be formulated as follows: For each mark $m$, we first predict $p^*(m)$, the probability that the mark of the next event is $m$, from $p^*(m, t)$. The expression of $p^*(m)$ is:

$$p^*(m) = \int_{t_l}^{+\infty} p^*(m, \tau) d\tau \tag{3}$$

The frequent mark has a high $p^*(m)$ and the rare mark has a low $p^*(m)$. Let $p^*(t|m)$ be the PDF on the time of the next event on the condition its mark is $m$. Based on $p^*(t|m)$, we can assume the mark of the next event is $m$ and estimate the expected time of the next event:

$$\bar{t}_m = \mathbb{E}_{t \sim p^*(\tau|m)}[t] = \int_{t_l}^{+\infty} \tau p^*(\tau|m) d\tau \tag{4}$$

$\bar{t}_m$ is the expected time of the next event on the condition that the mark of the next event is $m$. The result of RM-NEP is $\{(p^*(m), \bar{t}_m)\}_{m \in M}$.

While related, RM-NEP and NEP are different problems. NEP requires a single pair of $(m, t)$ as the predicted mark and time of the next event. Since only one mark is returned by NEP, some rare marks may never have a chance to be predicted as the next event. In this sense, the rare mark missing issue is intrinsic to NEP and irrelevant to the methods solving NEP. In contrast, RM-NEP returns $\{(p^*(m), \bar{t}_m)\}_{m \in M}$ so that rare marks have the equal chance as frequent marks in next event prediction.

## 3 METHODOLOGY

This section introduces our solution for RM-NEP. By the definition in Equation (3) and Equation (4), solving RM-NEP involves the improper integration of $p^*(m, \tau)$ and $\tau p^*(\tau|m)$, respectively, for each mark $m$. In general, improper integration does not have analytic solutions. This means directly calculating $p^*(m)$ and $\bar{t}_m$ is impossible. The solution is to approximate $p^*(m)$, and to approximate $\bar{t}$ by the average of $N$ samples $\{t^i\}_N^m$ from $p^*(t|m)$ as Equation (5).

$$\bar{t}_m = \mathbb{E}_{t \sim p^*(\tau|m)}[t] \approx \frac{1}{N} \sum_{i=1}^{N} t^i \tag{5}$$

### 3.1 UNIFYING INTEGRAL FUNCTIONS

The solution of RM-NEP involves the improper integration of two different functions in Equation (3) and Equation (4), respectively. Separately solving each integration problem is computationally inefficient. To address the challenge, we transform the improper integration of two different functions into one for an effective solution.

Because the improper integration in Equation (4) does not have analytic solutions, directly calculating $\bar{t}_m$ is impossible. A viable way is to estimate it by the average of $N$ samples $\{t^i\}_N^m$ from $p^*(t|m)$ as shown in Equation (5). To draw $\{t^i\}_N^m$ from $p^*(t|m)$, we use Inverse Transform Sampling (ITS), which takes the Cumulative Distribution Function (CDF) of the distribution that one wants to sample from. In our case, let $F^*(t|m)$ be the CDF of $p^*(t|m)$, i.e., $F^*(t|m) = \int_{t_l}^{t} p^*(\tau|m)d\tau$. $F^*(t|m)$ refers to the probability of the next event happening in $(t_l, t]$ on the condition that its mark is $m$. To draw a sample $t^i$ from $p^*(t|m)$, we need to solve Equation (6).

$$F^*(t^i|m) = u^i \tag{6}$$

where $u^i$ is a random sample from a uniform distribution $U(0, I)$. Since $F^*(t|m)$ is monotonic, Equation (6) is solvable by the bisection method. For each mark $m$, we obtain $\{t^i\}_N^m$ by solving Equation (6) $N$ times, which allows acquiring arbitrary number of samples for time prediction no matter rare or frequent the mark $m$ is. We can express $F^*(t|m)$ as follows:

$$F^*(t|m) = \frac{F^*(m, t)}{p^*(m)} = \frac{1}{\int_{t_l}^{+\infty} p^*(m, \tau)d\tau} \int_{t_l}^{t} p^*(m, \tau)d\tau \tag{7}$$

where $p^*(m) = \int_{t_l}^{+\infty} p^*(m, \tau)d\tau$ is the probability that the mark of next event is $m$ since $t_l$, and $F^*(m, t) = \int_{t_l}^{t} p^*(m, \tau)d\tau$ is the probability that the next event is mark $m$ and happens in time interval $(t_l, t]$. We can further breakdown $F^*(m, t)$ as shown Equation (8).

$$\begin{aligned} F^*(m, t) = \int_{t_l}^{t} p^*(m, \tau)d\tau = \int_{t_l}^{+\infty} p^*(m, \tau)d\tau - \int_{t}^{+\infty} p^*(m, \tau)d\tau \\ = \Gamma^*(m, t_l) - \Gamma^*(m, t) \end{aligned} \tag{8}$$

For each mark $m \in \mathrm{M}$, $\Gamma^*(m, t)$ is the integration starting from time $t$, any time after $t_l$ or $t_l$, to positive infinity. $\Gamma^*(m, t)$ is monotonically decreasing as its derivative $-p^*(m, t)$ is always smaller than 0. By definition, $p^*(m)$ in Equation (3) is equivalent to $\Gamma^*(m, t_l)$. That is, if we can solve $\Gamma^*(m, t)$, $p^*(m)$ can be solved by setting $t = t_l$. It means two different target integrals in Equation (3) and Equation (4) are now unified into one, i.e., $\Gamma^*(m, t)$.

While sampling a set of times from a distribution can follow Thinning Algorithm (TA) or Inverse Transform Sampling (ITS)(Rasmussen, 2018), only ITS is suitable for integral function unification here. The basic idea in ITS is to simulate using CDF of $p^*(t|m)$. Instead, Thinning Algorithm (TA) explicitly requires the expression of $p^*(t|m)$, which is unknown typically.

### 3.2 INTEGRAL-FREE NEURAL MARKED TEMPORAL POINT PROCESS (IFNMTPP)

Adopting $\Gamma^*(m, t)$ simplifies the procedure of calculating $p^*(m)$ and $\bar{t}_m$. However, $\Gamma^*(m, t)$ is an improper integration with infinitely long integration interval. In contrast, numeric integration

methods that most CIF-based MTPP models use are computationally heavy and can only estimate integrals on a finite interval. To effectively solve $\Gamma^*(m, t)$, this section introduces Integral-free Neural Marked Temporal Point Process (IFNMTPP). For each mark $m \in \mathrm{M}$, IFNMTPP models the relationship between $p^*(m, t)$ and its integral $\Gamma^*(m, t)$. IFNMTPP is inspired by, but different from, FullyNN(Omi et al., 2019) that models the relationship between $\lambda^*(m, t)$ and its integral[3].

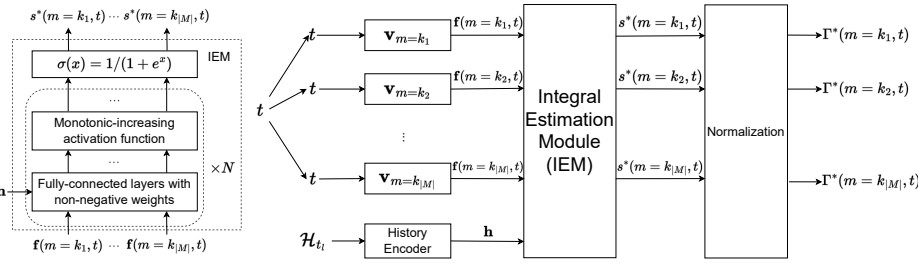

Figure 2: Architecture of IFNMTPP. The solid arrows refer to forward propagation The history encoder is an LSTM.

Figure 2 sketches the architecture of IFNMTPP. For each mark $m \in \mathrm{M}$, we assign a vector $\mathbf{v}_m$ to prepare $\mathbf{f}(m, t) = \mathbf{v}_m(t - t_l) + \mathbf{b}_m$ as input of the Integral Estimation Module (IEM). All parameters in $\mathbf{v}_m$ are non-negative. IEM contains multiple fully-connected layers with non-negative weights and monotonic-increasing activation functions. It ends with a monotonically decreasing function $\sigma(x) = 1/(1 + e^x)$ for each mark, so IFNMTPP is intrinsically monotonically decreasing w.r.t. $t$, matching the feature of $\Gamma^*(m, t)$. The outputs of IEM are scores $s^*(m = k_1, t), s^*(m = k_2, t), \cdots, s^*(m = k_{|\mathrm{M}|}, t)$. The value of $\sum_{m \in \mathrm{M}} s^*(m, t)$ is not guaranteed to be 1. To produce the qualified probability distribution, they need to be normalized. This is achieved by the Normalization module in Figure 2 that divides $s^*(m, t)$ by the partition function $Z(\mathcal{H}_{t_l}) = \sum_{m \in \mathrm{M}} s^*(m, t_l)$ for each $m \in M$. Finally, IFNMTPP outputs $\Gamma^*(m, t)$ for each mark $m$ at the given time $t$:

$$\Gamma^*(m, t) = \frac{s^*(m, t)}{Z(\mathcal{H}_{t_l})} \tag{9}$$

With $\Gamma^*(m, t)$ and $p^*(m)$, we have $F^*(t|m)$ by Equation (7) and Equation (8). Next, we calculate $\bar{t}_m$ by drawing $\{t^i\}_N^m$ from $F^*(t|m)$ following Equation (6). The loss function of IFNMTPP is:

$$L = - \sum_{(m_i, t_i) \in \mathcal{S}} \log p^*(m_i, t_i) - \log(\sum_{m \in M} \Gamma^*(m, T)). \tag{10}$$

where $p^*(m_i, t_i)$ is the predicted probability after $(i - 1)$th event, and $\log(\sum_{m \in M} \Gamma^*(m, T))$ is the survival term. In IFNMTPP, the expression of $p^*(m_i, t_i)$ is:

$$p^*(m_i, t_i) = -\frac{1}{Z(\mathcal{H}_{t_l})} \frac{\partial \Gamma^*(m_i, t_i)}{\partial s^*(m_i, t_i)} \frac{\partial s^*(m_i, t_i)}{\partial \mathbf{f}(m_i, t_i)} \frac{\partial \mathbf{f}(m_i, t_i)}{\partial t_i} \tag{11}$$

# 4 EXPERIMENTS

**Datasets**[4] Six real-world datasets include BookOrder(BO)(Du et al., 2016), Retweet(Zhao et al., 2015), StackOverflow(SO)(Leskovec & Krevl, 2014), Taobao User Behavior Data(Taobao)(Alibaba, 2018), Yelp[5], and earthquake events over the Conterminous US(USearthquake)(Xue et al., 2023). We split all marks of each dataset into two subsets, one containing frequent marks, denoted as $\mathrm{M}_f$, the other containing rare marks, denoted as $\mathrm{M}_r$. $\mathrm{M}_r \cap \mathrm{M}_f = \varnothing$ and $\mathrm{M}_r \cup \mathrm{M}_f = \mathrm{M}$. Marks in $\mathrm{M}_r$ have

---

[3]FullyNN considers the time point process (TPP) of events without mark which can be considered as MTPP with the same mark, i.e., $m$.

[4]BookOrder, Retweet, StackOverflow, Taobao, and USearthquake are released under Apache-2.0 license(Xue et al., 2023). Yelp is released under private license, but the license allows academic use.

[5]https://www.yelp.com/dataset

frequency lower than $5\%$ in StackOverflow, lower than $30\%$ in Yelp, lower than $50\%$ in BookOrder, and lower than $10\%$ in Taobao, Retweet, and USearthquake. We set different rare marks thresholds for different datasets based on their mark distribution, which can be found in Appendix C.5. Five synthetic datasets include Hawkes_1, Hawkes_2, Poisson, Self-correct, and Stationary Renewal(Omi et al., 2019). Details of these datasets are available in Appendix C.1.

**Baseline Models**[6] IFNMTPP is benchmarked against baselines. Among the state-of-the-art CIF-based MTPP models, four can be adapted to solve RM-NEP as baselines and we explain why the others cannot in Appendix C.4. The four CIF-based MTPP models are FullyNN(Omi et al., 2019), FENN[7], THP(Zuo et al., 2020), and SAHP(Zhang et al., 2020). These baselines use their own methods to model $p^*(m, t)$. Next, they approximate $\Gamma^*(m, t)$ using the numerical method by limiting the upper bound of the integration interval from infinity to a very large number. For a fair comparison, baselines estimate one improper integration $\Gamma^*(m, t)$ to predict $p^*(m)$ and $\bar{t}_m$ as our method in Section 3.1. Besides these four, another baseline Marked-LNM(Waghmare et al., 2022) models $p^*(m)$ using a classifier to predict the mark of the next event and models $p^*(t|m)$ using LogNormMix to predict the time of the event. More details of these baselines are available in Appendix C.4.

We do not include the intensity-free MTPP model(Shchur et al., 2020) and RMTPP(Recurrent Marked Temporal Point Process)(Du et al., 2016) because it does not model $p^*(m, t)$ but model $p^*(m)$ and $p^*(t)$ separately, and thus not suitable for RM-NEP. Also, we do not consider DDPM-based MTPP model(Yuan et al., 2023; Lüdke et al., 2023) because it directly generates the time and mark of the next event simultaneously by training a DDPM to enable a process to sample $p^*(m, t)$ from a normal distribution, and does not work for the particular mark as needed in RM-NEP.

**Evaluation Metrics** For synthetic datasets, because the true distribution $\hat{p}^*(m, t)$ is known, we can compare the learned $p^*(m, t)$ against the real one to evaluate the fidelity of a MTPP model. Most papers report the relative NLL loss, the average of the absolute difference between $-\log \hat{p}^*(m, t)$ and $-\log p^*(m, t)$ on the observed events(if markers are unavailable, $-\log \hat{p}^*(t)$ and $-\log p^*(t)$(Omi et al., 2019; Shchur et al., 2020)). The lower relative NLL loss indicates a better performance. However, such a metric only evaluates performance at discrete events, which cannot gauge the overall discrepancy between $\hat{p}^*(m, t)$ and $p^*(m, t)$. Therefore, we select Spearman Coefficient, $L^1$ distance and the relative NLL loss to measure the discrepancy between $\hat{p}^*(m, t)$ and $p^*(m, t)$ over time. Details of these metrics are available in Appendix C.3.

To measure the performance of RM-NEP solutions on real-world datasets, we use macro-F1 and MMAE (Mark-wise MAE). MMAE is a variant of MAE with consideration of marks. The test dataset $T$ contains many real next events. We denote $T_{m=k_i} \subset T$ as those real next events where the mark is $k_i \in M$. The number of events in $T_{m=k_i}$ is $|T_{m=k_i}|$. For each real next event $(m = k_i, t) \in T_{m=k_i}$, the task of RM-NEP is to predict $p^*(m)$ and time $\bar{t}_{m=k_i}$ if the next event has mark $m$. Here, we are interested in evaluating the predicted time. The absolute difference between $t$ and $\bar{t}_{m=k_i}$, $|t - \bar{t}_{m=k_i}|$, is the prediction error for the real next event $(m = k_i, t)$. Consider all real next events in $T_{m=k_i}$, $\text{MMAE}_{m=k_i}$ can be defined:

$$\text{MMAE}_{m=k_i} = \frac{1}{|T_{m=k_i}|} \sum_{(m=k_i, t) \in T_{m=k_i}} |t - \bar{t}_{m=k_i}| \qquad (12)$$

$\text{MMAE}_{M_*}$ is the geometric mean of $\text{MMAE}_{m=k_i}$ across all marks in $M_*$. $M_*$ can be M, $M_f$, or $M_r$:

$$\text{MMAE}_{M_*} = \sqrt[|M_*|]{\prod_{k_i \in M_*} \text{MMAE}_{m=k_i}} \qquad (13)$$

where $|M_*|$ is the number of marks in $M_*$. We also report macro-F1 on marks in M, $M_f$, and $M_r$. We run every experiment 3 times with different random seeds and report the mean and standard deviation (1-sigma) of all results.

### 4.1 PERFORMANCE OF IFNMTPP FOR TIME PREDICTION

The performance of IFNMTPP and baselines are reported in Table 1. First, we observe that $\text{MMAE}_{M_f}$ usually outperforms $\text{MMAE}_{M_r}$ across different MTPP approaches and datasets, except for BookOrder.

---

[6]Our codes will be released under MIT license.

[7]A variant FullyNN for handling marks, see Appendix B

Table 1: Time prediction performance on real-world datasets measured by MMAE, lower is better. The bold and underline indicate the best and the second-best values, respectively.

| | | BO | Retweet | SO | Taobao | USearthquake | Yelp |
|---|---|---|---|---|---|---|---|
| IFNMTPP (Ours) | $\text{MMAE}_M$ | **1.1967**±0.0076 | **2515.1**±6.5029 | **0.5212**±0.0142 | 0.3324±0.0579 | **0.6856**±0.0063 | 5.3271±0.0066 |
| | $\text{MMAE}_{M_r}$ | **1.1211**±0.0120 | 3291.2±29.097 | **0.4986**±0.0175 | 0.3385±0.0627 | **0.6966**±0.0081 | 5.3888±0.0078 |
| | $\text{MMAE}_{M_f}$ | **1.2775**±0.0093 | 2198.7±2.4798 | **0.6063**±0.0001 | 0.2529±0.0055 | **0.6713**±0.0048 | 5.2059±0.0071 |
| FENN | $\text{MMAE}_M$ | 124.15±1.1459 | 4449.5±20.281 | 1.0078±0.1321 | 2.0505±0.0579 | 0.8498±0.0567 | 5.3398±0.0045 |
| | $\text{MMAE}_{M_r}$ | 124.08±1.2262 | 6561.3±554.04 | 1.1430±0.1922 | 2.0648±0.4467 | 0.8422±0.0886 | 5.3587±0.0057 |
| | $\text{MMAE}_{M_f}$ | 124.20±1.1011 | 3674.7±173.54 | 0.6632±0.0062 | 1.8393±0.2966 | 0.8622±0.0291 | 5.3024±0.0041 |
| FullyNN | $\text{MMAE}_M$ | 125.70±0.9306 | 4704.4±104.82 | 0.6907±0.0076 | 2.4075±0.1788 | 0.9212±0.1057 | **5.3023**±0.0029 |
| | $\text{MMAE}_{M_r}$ | 125.42±0.9863 | 6985.0±224.54 | 0.7098±0.0094 | 2.4146±0.1858 | 0.9791±0.1132 | **5.3541**±0.0068 |
| | $\text{MMAE}_{M_f}$ | 125.98±0.8747 | 3860.9±67.116 | 0.6299±0.0148 | 2.2993±0.0746 | 0.8495±0.0977 | **5.2001**±0.0047 |
| SAHP | $\text{MMAE}_M$ | 5.3994±0.0329 | 3387.4±144.84 | 0.7974±0.0538 | 1.0461±0.1901 | 0.7317±0.0244 | 5.3174±0.0172 |
| | $\text{MMAE}_{M_r}$ | 3.0312±0.4705 | 5010.0±575.18 | 0.7827±0.0657 | 1.1346±0.2164 | 0.7474±0.0278 | 5.3674±0.0206 |
| | $\text{MMAE}_{M_f}$ | 9.8680±3.5641 | 2791.1±15.472 | 0.8515±0.0108 | 0.2888±0.0174 | 0.7112±0.0204 | 5.2187±0.0114 |
| THP | $\text{MMAE}_M$ | 2.0856±0.5256 | 4096.7±444.75 | 0.6750±0.0138 | 2.7307±0.5392 | 0.9488±0.0148 | 5.3706±0.0392 |
| | $\text{MMAE}_{M_r}$ | 1.9560±0.5065 | 4701.3±441.66 | 0.6909±0.0152 | 2.6824±0.5545 | 0.9680±0.0072 | 5.4238±0.0424 |
| | $\text{MMAE}_{M_f}$ | 2.2246±0.5474 | 3824.6±443.47 | 0.6238±0.0100 | 3.6884±0.1286 | 0.9241±0.0270 | 5.2657±0.0332 |
| Marked-LNM | $\text{MMAE}_M$ | 1.7400±0.5093 | 2559.8±5.9380 | 0.9067±0.3687 | **0.2058**±0.0079 | 0.7646±0.0026 | 5.3291±0.0046 |
| | $\text{MMAE}_{M_r}$ | 2.1514±1.0044 | 3314.3±1.2460 | 1.0520±0.5330 | **0.2043**±0.0091 | 0.7773±0.0057 | 5.3783±0.0032 |
| | $\text{MMAE}_{M_f}$ | 1.4618±0.1414 | 2249.7±7.4050 | 0.6084±0.0007 | **0.2318**±0.0128 | 0.7480±0.0013 | 5.2322±0.0072 |

Table 2: The evaluation time measured in seconds on the test datasets. Lower is better.

| | IFNMTPP (Ours) | FENN | FullyNN | SAHP | THP | Marked-LNM |
|---|---|---|---|---|---|---|
| BO | **10.104** | 362.01 | 356.82 | 136.82 | 54.305 | 16.859 |
| Retweet | **255.40** | 6490.6 | 6312.4 | 1838.6 | 1682.4 | 329.29 |
| SO | **140.69** | 27078 | 26638 | 956.49 | 1252.9 | 797.77 |
| Taobao | **79.610** | 25810 | 25237 | 1238.6 | 770.35 | 279.42 |
| USearthquake | **86.374** | 476.58 | 469.95 | 254.02 | 262.26 | 185.44 |
| Yelp | **38.678** | 703.05 | 680.99 | 176.53 | 161.17 | 51.630 |

This is expected since frequent marks have more training data than rare marks, leading to a more accurate estimation of $p^*(m,t)$ for frequent marks. This finding does not fit the BookOrder. One possible reason is that the mark distribution of BookOrder is balanced. Second, compared with CIF-based baselines, IFNMTPP demonstrates superior performances in most cases. As discussed in Section 3.1, the time prediction for each mark $m$ is sampled from $p^*(t|m)$ based on the values of $\Gamma^*(m,t)$ at many different times. The accurate approximation of $\Gamma^*(m,t)$ leads to accurate time prediction. Besides, IFNMTPP also outperforms Marked-LNM. This demonstrates that modeling $\Gamma^*(m,t)$ is better than directly modeling $p^*(t|m)$ by the composition of log-normal distributions.

Table 2 reports the evaluation time of all MTPP models. Compared with CIF baselines, IFNMTPP is faster by 1-2 orders of magnitude. Section 3.2 tells that the CIF-based baselines are computationally heavy because they use the numerical method to calculate $\Gamma^*(m,t)$. In contrast, IFNMTPP directly models $\Gamma^*(m,t)$, which is more straightforward with low and consistent computation cost. Marked-LNM does not involve integral estimation during modeling and drawing samples from $p^*(m)$ and $p^*(t|m)$. So, it is faster than all CIF-based baselines and should be comparable with our IFNMTPP. We reckon that the speed difference between Marked-LNM and IFNMTPP is due to implementation. These experiment results focus on the accuracy and efficiency of calculating $p^*(t|m)$ and its integral $F^*(t|m)$. IFNMTPP also has strong performances on modeling $p^*(m,t)$ and $p^*(m)$ against other baselines. These results are available in Appendix D.

## 4.2 PERFORMANCE OF IFNMTPP FOR MARK PREDICTION

The accuracy of the mark predicted by $p^*(m)$ from IFNMTPP and baselines are reported in Table 3. The metric is macro-F1. The higher macro-F1 indicates more marks are predicted correctly. For calculating macro-F1, the mark with the highest $p^*(m)$ is selected as the mark prediction. $p^*(m)$ is

the value of $\Gamma^*(m,t)$ at a single time $t = t_l$. More accurate $\Gamma^*(m,t)$ should lead to more accurate $p^*(m)$, which can help correctly predict the mark. However, the accuracy improvement on $p^*(m)$ is limited compared with that on $\bar{t}_m$. The reason is the mark prediction only involves the value of $\Gamma^*(m,t)$ at a single time $t = t_l$, while $\bar{t}_m$ is based on the value of $\Gamma^*(m,t)$ at many different times.

Table 3: Mark prediction performance, measured by macro-F1, on real-world datasets. Higher is better. The bold and underline indicate the best and the second-best values, respectively.

| | | BO | Retweet | SO | Taobao | USearthquake | Yelp |
|---|---|---|---|---|---|---|---|
| IFNMTPP (Ours) | All Marks | $\mathbf{0.6003}_{\pm 0.0009}$ | $0.3569_{\pm 0.0001}$ | $\mathbf{0.1519}_{\pm 0.0033}$ | $\mathbf{0.2338}_{\pm 0.0258}$ | $0.1795_{\pm 0.0078}$ | $0.2524_{\pm 0.0009}$ |
| | Rare Marks | $0.7235_{\pm 0.0032}$ | $\mathbf{0.0014}_{\pm 0.0002}$ | $0.1457_{\pm 0.0076}$ | $\mathbf{0.1324}_{\pm 0.0054}$ | $0.0012_{\pm 0.0008}$ | $0.0376_{\pm 0.0019}$ |
| | Frequent Marks | $0.7573_{\pm 0.0043}$ | $0.5057_{\pm 0.0004}$ | $0.1364_{\pm 0.0009}$ | $0.3186_{\pm 0.0541}$ | $\underline{0.2525}_{\pm 0.0110}$ | $\underline{0.7986}_{\pm 0.0107}$ |
| FENN | All Marks | $0.3923_{\pm 0.0580}$ | $\mathbf{0.3673}_{\pm 0.0007}$ | $0.0938_{\pm 0.0002}$ | $0.1283_{\pm 0.0104}$ | $\mathbf{0.1835}_{\pm 0.0079}$ | $0.2436_{\pm 0.0029}$ |
| | Rare Marks | $0.0408_{\pm 0.0062}$ | $\underline{0.0013}_{\pm 0.0000}$ | $0.0298_{\pm 0.0015}$ | $0.0210_{\pm 0.0129}$ | $0.0006_{\pm 0.0004}$ | $0.0160_{\pm 0.0066}$ |
| | Frequent Marks | $\underline{0.9885}_{\pm 0.0031}$ | $\mathbf{0.5195}_{\pm 0.0015}$ | $\mathbf{0.1512}_{\pm 0.0003}$ | $\underline{0.4252}_{\pm 0.2559}$ | $\mathbf{0.2587}_{\pm 0.0101}$ | $\mathbf{0.8540}_{\pm 0.0754}$ |
| FullyNN | All Marks | $0.3339_{\pm 0.0000}$ | $0.2316_{\pm 0.0000}$ | $0.0121_{\pm 0.0000}$ | $0.0194_{\pm 0.0000}$ | $0.1621_{\pm 0.0000}$ | $0.0953_{\pm 0.0000}$ |
| | Rare Marks | $0.0000_{\pm 0.0000}$ | $0.0000_{\pm 0.0000}$ | $0.0000_{\pm 0.0000}$ | $\underline{0.0437}_{\pm 0.0000}$ | $0.0000_{\pm 0.0000}$ | $\mathbf{0.2634}_{\pm 0.0000}$ |
| | Frequent Marks | $\mathbf{1.0000}_{\pm 0.0000}$ | $0.3282_{\pm 0.0000}$ | $0.0287_{\pm 0.0000}$ | $0.0000_{\pm 0.0000}$ | $0.2316_{\pm 0.0000}$ | $0.0000_{\pm 0.0000}$ |
| SAHP | All Marks | $0.5987_{\pm 0.0021}$ | $\underline{0.3588}_{\pm 0.0007}$ | $0.1378_{\pm 0.0038}$ | $0.1487_{\pm 0.0111}$ | $0.1637_{\pm 0.0005}$ | $\mathbf{0.2564}_{\pm 0.0034}$ |
| | Rare Marks | $0.7177_{\pm 0.0156}$ | $\underline{0.0013}_{\pm 0.0021}$ | $0.1184_{\pm 0.0086}$ | $0.0094_{\pm 0.0113}$ | $0.0007_{\pm 0.0001}$ | $0.0446_{\pm 0.0014}$ |
| | Frequent Marks | $0.7586_{\pm 0.0145}$ | $\underline{0.5082}_{\pm 0.0012}$ | $0.1427_{\pm 0.0012}$ | $\mathbf{0.9446}_{\pm 0.0682}$ | $0.2327_{\pm 0.0007}$ | $0.7056_{\pm 0.0717}$ |
| THP | All Marks | $0.3904_{\pm 0.0804}$ | $0.2365_{\pm 0.0069}$ | $0.0965_{\pm 0.0096}$ | $0.0100_{\pm 0.0045}$ | $0.1627_{\pm 0.0004}$ | $0.2534_{\pm 0.0034}$ |
| | Rare Marks | $\mathbf{0.9966}_{\pm 0.0004}$ | $0.0000_{\pm 0.0000}$ | $0.0415_{\pm 0.0266}$ | $0.0202_{\pm 0.0087}$ | $0.0000_{\pm 0.0000}$ | $0.0336_{\pm 0.0099}$ |
| | Frequent Marks | $0.0131_{\pm 0.0004}$ | $0.3350_{\pm 0.0097}$ | $0.1415_{\pm 0.0215}$ | $0.0000_{\pm 0.0000}$ | $0.2323_{\pm 0.0005}$ | $0.6561_{\pm 0.0375}$ |
| Marked-LNM | All Marks | $\mathbf{0.6003}_{\pm 0.0026}$ | $0.3566_{\pm 0.0018}$ | $0.1484_{\pm 0.0009}$ | $0.1729_{\pm 0.0302}$ | $0.1674_{\pm 0.0014}$ | $\underline{0.2538}_{\pm 0.0021}$ |
| | Rare Marks | $0.7404_{\pm 0.0030}$ | $0.0012_{\pm 0.0002}$ | $\mathbf{0.1506}_{\pm 0.0025}$ | $0.0378_{\pm 0.0343}$ | $0.0006_{\pm 0.0003}$ | $\underline{0.0397}_{\pm 0.0050}$ |
| | Frequent Marks | $0.7510_{\pm 0.0055}$ | $0.5053_{\pm 0.0028}$ | $0.1297_{\pm 0.0003}$ | $0.6483_{\pm 0.2459}$ | $0.2386_{\pm 0.0014}$ | $\underline{0.7765}_{\pm 0.0202}$ |

## 4.3 EVALUATING MODEL FIDELITY ON SYNTHETIC DATASETS

On five synthetic datasets where real $p^*(m,t)$ is known, IFNMTPP and baselines are compared in their ability of $p^*(m,t)$ modeling in Table 4. We select the Spearman coefficient, $L^1$ distance, and the relative NLL loss to gauge the difference between the learned $p^*(m,t)$ and the ground truth distribution $\hat{p}^*(m,t)$. Details of evaluation metrics are available in Appendix C.3. The result shows that IFNMTPP consistently learns more accurate distributions than all baselines.

Table 4: Model fidelity test performance on synthetic datasets; higher Spearman, lower $L^1$ and relative NLL loss are better; the bold and underline indicate the best and the second-best values, respectively.

| | | Hawkes_1 | Hawkes_2 | Poisson | Self-correct | Stationary Renewal |
|---|---|---|---|---|---|---|
| Spearman | IFNMTPP (Ours) | $\mathbf{1.0000}_{\pm 0.0000}$ | $\mathbf{0.9999}_{\pm 0.0000}$ | $\mathbf{1.0000}_{\pm 0.0000}$ | $\mathbf{0.9551}_{\pm 0.0009}$ | $\mathbf{0.9999}_{\pm 0.0000}$ |
| | FENN | $0.9946_{\pm 0.0004}$ | $\underline{0.9964}_{\pm 0.0002}$ | $0.9736_{\pm 0.0006}$ | $0.9473_{\pm 0.0010}$ | $\underline{0.9998}_{\pm 0.0000}$ |
| | FullyNN | $0.9952_{\pm 0.0004}$ | $0.9963_{\pm 0.0002}$ | $0.9722_{\pm 0.0018}$ | $0.9477_{\pm 0.0001}$ | $\underline{0.9998}_{\pm 0.0000}$ |
| | SAHP | $\underline{0.9959}_{\pm 0.0047}$ | $0.9862_{\pm 0.0000}$ | $0.9615_{\pm 0.0025}$ | $\underline{0.9492}_{\pm 0.0014}$ | $0.9990_{\pm 0.0007}$ |
| | THP | $0.9266_{\pm 0.0026}$ | $0.7366_{\pm 0.0005}$ | $\mathbf{1.0000}_{\pm 0.0000}$ | $0.6969_{\pm 0.0017}$ | $0.0413_{\pm 0.0024}$ |
| | Marked-LNM | $0.9924_{\pm 0.0007}$ | $0.9971_{\pm 0.0001}$ | $0.9713_{\pm 0.0024}$ | $0.9491_{\pm 0.0005}$ | $\mathbf{0.9999}_{\pm 0.0000}$ |
| $L^1$ | IFNMTPP (Ours) | $\mathbf{0.1480}_{\pm 0.0085}$ | $\mathbf{0.3105}_{\pm 0.0432}$ | $\mathbf{0.0133}_{\pm 0.0091}$ | $\mathbf{0.5163}_{\pm 0.0290}$ | $\underline{0.0654}_{\pm 0.0018}$ |
| | FENN | $0.6248_{\pm 0.0052}$ | $3.0398_{\pm 0.0693}$ | $0.2919_{\pm 0.0051}$ | $1.2139_{\pm 0.1652}$ | $0.0703_{\pm 0.0058}$ |
| | FullyNN | $\underline{0.6235}_{\pm 0.0227}$ | $3.1048_{\pm 0.0763}$ | $0.2973_{\pm 0.0098}$ | $1.1889_{\pm 0.0244}$ | $0.0710_{\pm 0.0099}$ |
| | SAHP | $1.0245_{\pm 0.2967}$ | $4.7867_{\pm 0.2735}$ | $0.6893_{\pm 0.0238}$ | $1.3363_{\pm 0.0196}$ | $0.4872_{\pm 0.1833}$ |
| | THP | $12.003_{\pm 0.2069}$ | $25.500_{\pm 0.3642}$ | $\underline{0.0203}_{\pm 0.0067}$ | $10.656_{\pm 0.0965}$ | $9.9230_{\pm 0.0451}$ |
| | Marked-LNM | $0.6994_{\pm 0.0117}$ | $2.6446_{\pm 0.0633}$ | $0.3620_{\pm 0.0044}$ | $\underline{0.7406}_{\pm 0.0168}$ | $\mathbf{0.0402}_{\pm 0.0001}$ |
| Relative NLL | IFNMTPP (Ours) | $\mathbf{0.0000}_{\pm 0.0000}$ | $\mathbf{0.0001}_{\pm 0.0000}$ | $\mathbf{0.0000}_{\pm 0.0000}$ | $\mathbf{0.0007}_{\pm 0.0003}$ | $\mathbf{0.0000}_{\pm 0.0000}$ |
| | FENN | $\underline{0.0003}_{\pm 0.0000}$ | $0.0009_{\pm 0.0001}$ | $0.0002_{\pm 0.0000}$ | $0.0016_{\pm 0.0006}$ | $\mathbf{0.0000}_{\pm 0.0000}$ |
| | FullyNN | $\underline{0.0003}_{\pm 0.0000}$ | $\underline{0.0008}_{\pm 0.0001}$ | $0.0002_{\pm 0.0000}$ | $\underline{0.0015}_{\pm 0.0001}$ | $\mathbf{0.0000}_{\pm 0.0000}$ |
| | SAHP | $0.0086_{\pm 0.0017}$ | $0.0312_{\pm 0.0193}$ | $0.0092_{\pm 0.0002}$ | $0.0072_{\pm 0.0009}$ | $\underline{0.0034}_{\pm 0.0010}$ |
| | THP | $0.2137_{\pm 0.0001}$ | $0.6663_{\pm 0.0029}$ | $\mathbf{0.0000}_{\pm 0.0000}$ | $0.1262_{\pm 0.0004}$ | $0.0771_{\pm 0.0000}$ |
| | Marked-LNM | $0.0004_{\pm 0.0000}$ | $0.0010_{\pm 0.0000}$ | $0.0006_{\pm 0.0000}$ | $0.0018_{\pm 0.0001}$ | $\mathbf{0.0000}_{\pm 0.0000}$ |

## 5 RELATED WORK

To solve NEP, most MTPP studies specify a separate Conditional Intensity Function (CIF) $\lambda^*(m, t)$ for each categorical mark $m$, based on which $p^*(m, t)$ can be formulated(Daley & Vere-Jones, 2003; Mei & Eisner, 2017; Zuo et al., 2020; Zhang et al., 2020; Enguehard et al., 2020; Mei et al., 2022). All these studies assume $\lambda^*(m, t)$ has a specific functional form which can be integrated to infer $p^*(m, t)$. As pointed out by Shchur et al. (2020), requirement of integration is the intrinsic shortcomings of CIF models due to the trade-off between efficiency and effectiveness. A more sophisticated intensity function(Mei & Eisner, 2017; Zuo et al., 2020; Zhang et al., 2020; Mei et al., 2022) can better capture the system dynamics but will require approximating the integral of $\lambda^*(m, t)$ using a numerical method such as Monte Carlo. Recurrent Marked Temporal Point Process(RMTPP)(Du et al., 2016) eludes numerical integral approximation as the CIF and its integral have a closed form, which makes the log-likelihood easy to compute. However, RMTPP ignores the relation between mark and time because of factorizing $p^*(m, t)$ into two independent distribution $p^*(t)$ and $p^*(m)$. Moreover, the predefined closed-form CIF usually has limited expressiveness.

Recent studies move away from directly modeling CIF. Shchur et al. (2020) proposed an intensity-free solution, called LogNormMix, to infer the density function $p^*(t)$ from a simple distribution such as the mixture of log-normal distributions. In the scenarios of multiple marks, the intensity-free solution factorizes $p^*(m, t)$ into a product of two independent distributions $p^*(t)$ and $p^*(m)$. Omi et al. (2019) proposed FullyNN to model the integral of CIF using a neural network where CIF can be derived by differentiation, an operation computationally much easier compared with integration. FullyNN was proposed for TPP rather than MTPP, which does not consider event marks. Also, FullyNN cannot guarantee essential mathematical restrictions(Shchur et al., 2020). The idea of FullyNN inspired integral-based Spatio-temporal Point Process (STPP) models(Zhang et al., 2023; Zhou & Yu, 2023), where the marks are locations in a continuous spatial space.

All MTPP studies discussed so far predict the time of the next event first and then predict the mark. Recently, Waghmare et al. (2022) proposes to model $p^*(m)$ using a classifier to predict the mark of the next event and modeling $p^*(t|m)$ to predict the time of the event based on LogNormMix. Besides the classic MTPP approaches, some researchers explore approaches that directly generate the time and mark of the next event simultaneously. Yuan et al. (2023) and Lüdke et al. (2023) suggest to represent the conditional distribution by a Denoising Diffusion Probabilistic Model (DDPM)(Ho et al., 2020). Lüdke et al. (2023) applied denoising diffusion to convert a long event sequence sampled from a predefined Poisson distribution to any given distribution. Meanwhile, Yuan et al. (2023) proposed DSTPP, which employs DDPM to represent STPP.

In summary, the existing studies in the current literature return a single mark and a single time as the prediction of the next event, i.e., NEP. None of them consider the imbalanced distribution of marks. The frequent marks dominate the results of NEP and rare marks are mostly missing in the next event prediction as shown in Figure 1. This is unacceptable in many application scenarios as discussed in Section 1. Our study aims to address this issue.

## 6 LIMITATION AND CONCLUSION

**Limitation** One limitation is the experiment results on efficiency are measured by the total evaluation time on the test datasets instead of the expected FLOP counting. However we carefully managed the experiment environment and every run has the same computation resources, so we believe that the evaluation time can properly demonstrate the efficiency of the models.

**Conclusion** In the existing MTPP studies, the results of Next-event Prediction problem (NEP) are dominated by frequent marks, and the rare marks may be never present. This situation is unacceptable in many applications if the rare mark is critical such as major earthquakes. To fill the gap, this study tackles the novel Rare-mark-aware Next Event Prediction problem (RM-NEP) to answer two questions for each mark $m$: "*what is the probability that the mark of the next event is $m$?* and "*if $m$, when will the next event happen?*". This study solves RM-NEP accurately and efficiently by unifying two different improper integration functions into $\Gamma^*(m, t)$ and developing a novel Integral-free Neural Marked Temporal Point Process (IFNMTPP) to directly calculate $\Gamma^*(m, t)$. Extensive experiments on real-world and synthetic datasets demonstrate the superior performance of our solution for RM-NEP against various baselines.

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

## A THE CONDITIONAL JOINT PDF

This study concerns events with categorical marks. For mark $m$, we define a conditional intensity function $\lambda^*(m, t)$:

$$
\begin{aligned}
\lambda^*(m = k_i, t) &= \lambda(m = k_i, t | \mathcal{H}_t) \\
&= \lim_{\Delta t \to 0} \frac{P(m = k_i, t \in [t, t+\Delta t) | \mathcal{H}_{t-})}{\Delta t} \\
&= \lim_{\Delta t \to 0} \frac{p(m = k_i, t \in [t, t+\Delta t) | \mathcal{H}_{t_l}) \Delta t}{P(\forall j \in \mathbb{N}^+, t_j \notin (t_l, t) | \mathcal{H}_{t_l}) \Delta t} \\
&= \lim_{\Delta t \to 0} \frac{p(m = k_i, t \in [t, t+\Delta t) | \mathcal{H}_{t_l})}{P(\forall j \in \mathbb{N}^+, t_j \notin (t_l, t) | \mathcal{H}_{t_l})} \\
&= \frac{p(m = k_i, t \in [t, t+dt) | \mathcal{H}_{t_l})}{P(\forall j \in \mathbb{N}^+, t_j \notin (t_l, t) | \mathcal{H}_{t_l})}
\end{aligned}
\tag{14}
$$

where $\mathcal{H}_{t_l}$ is the history up to (including) the most recent event, $\mathcal{H}_{t-}$ is the history up to (excluding) the current time, $P(\forall j \in \mathbb{N}^+, t_j \notin (t_l, t) | \mathcal{H}_{t_l})$ represents the probability that no event is observed in time interval $(t_l, t)$ given $\mathcal{H}_{t_l}$.

We denote $P'_m((t_1, t_2) | \mathcal{H}_{t_l})$ for the conditional probability that an event $m$ happens in $(t_1, t_2)$. Following the definition of simple TPP that at most one event happens at every timestamp $t$, the probability that no event occurs in $(t_l, t)$ is:

$$
\begin{aligned}
&P(\forall j \in \mathbb{N}^+, t_j \notin (t_l, t) | \mathcal{H}_{t_l}) \\
=& 1 - \sum_{m \in M} P'_m((t_l, t) | \mathcal{H}_{t_l}) \prod_{n \in M, n \neq m} (1 - P'_n((t_l, t) | \mathcal{H}_{t_l})) \\
=& 1 - \sum_{m \in M} \frac{P'_m((t_l, t) | \mathcal{H}_{t_l})}{1 - P'_m((t_l, t) | \mathcal{H}_{t_l})} \prod_{n \in M} (1 - P'_n((t_l, t) | \mathcal{H}_{t_l})) \\
=& 1 - \sum_{m \in M} F(m, t | \mathcal{H}_{t_l}) = 1 - \sum_{m \in M} F^*(m, t)
\end{aligned}
\tag{15}
$$

where

$$
F^*(m, t) = \frac{P'_m((t_l, t) | \mathcal{H}_{t_l})}{1 - P'_m((t_l, t) | \mathcal{H}_{t_l})} \prod_{n \in M} (1 - P'_n((t_l, t) | \mathcal{H}_{t_l}))
\tag{16}
$$

The conditional joint PDF that the next event is $m$ and occurs in $[t, t+dt)$ is:

$$
p(m = k_i, t \in [t, t+\Delta t) | \mathcal{H}_{t_l}) = \frac{dF^*(m = k_i, t)}{dt}
\tag{17a}
$$

$$
\int_{t_l}^{t} p(m = k_i, t \in [t, t+\Delta t) | \mathcal{H}_{t_l}) d\tau = F^*(m = k_i, t)
\tag{17b}
$$

In this study, $p^*(m, t)$, shorthand of $p(m, t | \mathcal{H}_{t_l})$, is the formal representation of $p(m = k_i, t \in [t, t+\Delta t) | \mathcal{H}_{t_l})$. Note $F^*(m, t)$ in Equation (16) is the probability that only one event happens in interval $[t, t+dt)$ and the mark is $m$. This ensures the MTPP represented by $p^*(m, t)$ is simple. By integrating Equation (17a) and Equation (15) in Equation (14), we have

$$
p^*(m, t) = \lambda^*(m, t)(1 - \sum_{w \in M} F^*(w, t))
\tag{18}
$$

where $\sum_{w \in M} F^*(w, t)$ is calculated from the sum of Equation (14) over marker $m$:

$$
\sum_{w \in M} F^*(w, t) = 1 - \exp(-\int_{t_l}^{t} \sum_{n \in M} \lambda^*(n, \tau) d\tau)
\tag{19}
$$

Then, we solve $p^*(m, t)$:

$$
p^*(m, t) = \lambda^*(m, t) \exp(-\int_{t_l}^{t} \sum_{n \in M} \lambda^*(n, \tau) d\tau)
\tag{20}
$$

which is equivalent with Equation (2).

# B   ANALYSIS ON FULLYNN AND FENN

## B.1   FULLYNN AND ITS SHORTCOMING

FullyNN sets the learning target of NNs to $\Lambda^*(t)$, the integral of intensity functions $\lambda^*(t)$. This idea works when no event mark information is present. However, when we require FullyNN to learn different intensity functions for different event marks, the limitation emerges as FullyNN cannot allocate different computation graphs for different marks.

To understand this, we should recognize how FullyNN calculates the intensity function. Following the definition in (Omi et al., 2019), a FullyNN can be written in the following expression:

$$\Lambda^*(t) = \text{FullyNN}(t, \mathbf{h}) = \text{IEM}(\mathbf{f}(t), \mathbf{h}) \tag{21}$$

where $\mathbf{f}(t)$ represents a monotonic-increasing function mapping the time $t$ into a vector, $\mathbf{h}$ is the history embedding, and IEM refers to the integral estimation module. From Equation (21), we could derive the intensity function as:

$$\lambda^*(t) = \frac{\partial \Lambda^*(t)}{\partial t} = \frac{\partial \text{IEM}(\mathbf{f}(t), \mathbf{h})}{\partial \mathbf{f}(t)} \frac{\partial \mathbf{f}(t)}{\partial t} \tag{22}$$

If one generalizes the FullyNN from mark-agnostic to mark-aware by simply expanding the input time from $t$ to $\mathbf{t} = [t, t, t, \cdots, t]^\top$, each for one of the $|\text{M}|$ marks, and they share the same vector $\mathbf{v}$ for generating the same $\mathbf{f}(t)$s as input of IEM. By letting the corresponding intensity integral be $\mathbf{\Lambda}^*(\mathbf{t}) = [\Lambda^*(m = k_1, t), \Lambda^*(m = k_2, t), \Lambda^*(m = k_3, t), \cdots, \Lambda^*(m = k_{|\text{M}|}, t)]^\top$, we could find the Jacobian Matrix $D_{\mathbf{t}} \mathbf{\Lambda}^*(t)$ is:

$$
\begin{aligned}
& D_{\mathbf{t}} \mathbf{\Lambda}^*(\mathbf{t}) \\
& = \frac{\partial [\Lambda^*(m = k_1, t), \Lambda^*(m = k_2, t), \cdots, \Lambda^*(m = k_{|M|}, t)]^\top}{\partial \mathbf{t}} \\
& = \begin{pmatrix}
\frac{\partial \text{IEM}(\mathbf{f}(t), \mathbf{h})}{\partial \mathbf{f}(t)} \mathbf{v} & 0 & \cdots & 0 \\
0 & \frac{\partial \text{IEM}(\mathbf{f}(t), \mathbf{h})}{\partial \mathbf{f}(t)} \mathbf{v} & \cdots & 0 \\
\vdots & \vdots & \ddots & \vdots \\
0 & 0 & \cdots & \frac{\partial \text{IEM}(\mathbf{f}(t), \mathbf{h})}{\partial \mathbf{f}(t)} \mathbf{v}
\end{pmatrix}
\end{aligned} \tag{23}
$$

which implies that the intensity functions for different marks receive identical distributions, and the event prediction performance would be stuck at $\frac{1}{|\text{M}|}$. We believe this might explain the shockingly bad event prediction performance in (Enguehard et al., 2020).

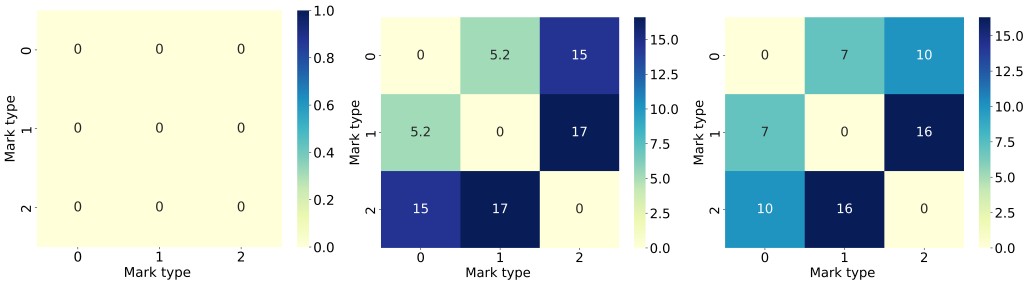

(a) FullyNN on Retweet dataset.     (b) FENN on Retweet dataset.     (c) IFNMTPP on Retweet dataset.

Figure 3: The $L^1$ distance between distribution $p^*(m_i, t)$ and distribution $p^*(m_j, t)$, for each pair of marks $(m_i, m_j)$ in M, generated by FullyNN, FENN and IFNMTPP on an event sequence in Retweet dataset. FullyNN generates the identical distribution for different marks as the $L^1$ distance between $p^*(m_i, t)$ and $p^*(m_j, t)$ is 0 for each pair of marks $(m_i, m_j)$ in M. In contrast, FENN and IFNMTPP generate different distributions for different marks.

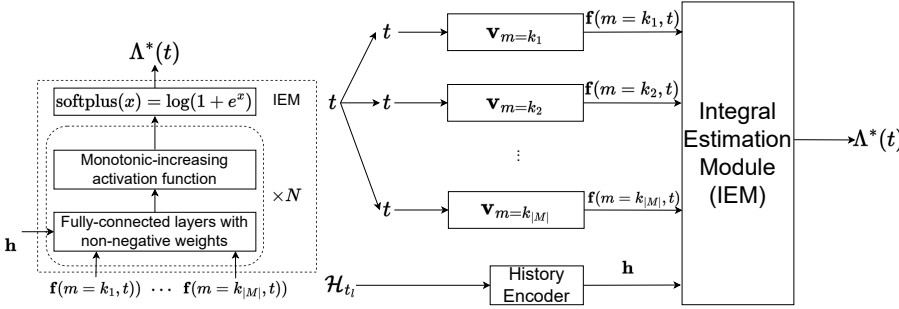

Figure 4: Architecture of Fully Event Neural Network (FENN). The solid arrows refer to forward propagation. FENN models $\Lambda^*(t)$ and obtains $\lambda^*(m, t)$ by backpropagation.

## B.2 FULLY EVENT NEURAL NETWORK

To handle marks in a better way, FullyNN can be extended to FENN (Fully Event Neural Network), which models $p^*(m, t)$ based on the conditional intensity $\lambda^*(m, t)$ as defined in Equation (2). FENN is sketched in Figure 4. The history $\mathcal{H}_{t_l}$ is represented as an embedding $\mathbf{h}$ using a LSTM encoder(Omi et al., 2019). FENN needs to model $|\mathrm{M}|$ conditional intensity functions, i.e., $\lambda^*(m, t)$ for all $m \in \mathrm{M}$. The integral of conditional intensity functions across all marks, from the time of the latest event $t_l$ to the current time $t$, is denoted as $\Lambda^*(t)$. The definition of $\Lambda^*(t)$ is given in Equation (24a), and the relationship between $\Lambda^*(t)$ and $\lambda^*(m, t)$ is presented in Equation (24b):

$$\Lambda^*(t) = \int_{t_l}^{t} \sum_{n \in \mathrm{M}} \lambda^*(n, \tau)d\tau = \mathrm{IEM}_{\mathrm{FENN}}(\mathbf{f}(m = k_1, t), t), \cdots, \mathbf{f}(m = k_{|\mathrm{M}|}, t), \mathbf{h}), \quad (24a)$$

$$\lambda^*(m, t) = \frac{\partial \Lambda^*(t)}{\partial \mathbf{f}(m, t)} \frac{\partial \mathbf{f}(m, t)}{\partial t}, \text{ for } m \in \mathrm{M}. \quad (24b)$$

where $\mathbf{f}(m, t)$ is defined as $\mathbf{v}_m(t - t_l)$ for mark $m \in \mathrm{M}$. FENN utilizes $\mathbf{v}_m$ to distinguish event marks to avoid sharing the same computation graph as FullyNN. The integral estimation module (IEM) of FENN contains multiple fully-connected layers with non-negative weights and monotonic-increasing activation functions and ends with an unbounded above softplus function $\mathrm{softplus}(x) = \log(1 + e^x)$. IEM receives history embedding $\mathbf{h}$ and $\mathbf{f}(m = k_1, t), \mathbf{f}(m = k_2, t), \cdots, \mathbf{f}(m = k_{|\mathrm{M}|}, t)$, outputs $\Lambda^*(t)$. The loss function of FENN is the negative logarithm of $p^*(m, t)$ at every known event $(m_i, t_i) \in \mathcal{S}$, as shown in Equation (25).

$$L = \sum_{(m_i, t_i) \in \mathcal{S}} -\log p^*(m_i, t_i)$$

$$= \sum_{(m_i, t_i) \in \mathcal{S}} \left( -\log \lambda^*(m_i, t_i) + \int_{t_l}^{t_i} \lambda^*(m, \tau)d\tau \right) \quad (25)$$

FENN inherits from FullyNN the capability to instantaneously provide accurate $\lambda^*(m, t)$ and the integral $\Lambda^*(t)$, and FENN solves the computation graph overlap issue in FullyNN as evidenced by the test results in Figure 3. However, FENN also inherits the weakness of FullyNN: it violates several essential mathematical restrictions(Shchur et al., 2020). To elaborate on this, we first rewrite Equation (24a) by expanding $\mathrm{IEM}_{\mathrm{FENN}}(\cdot)$.

$$\Lambda^*(t) = \sum_{m \in M} \mathrm{softplus}(\Omega^*(m, t) + b) \quad (26a)$$

$$\Omega^*(m, t) = \mathbf{w}^\top \tanh(\mathcal{F}_l(\cdots \mathcal{F}_2(\mathcal{F}_1([\mathbf{v}_m(t - t_l), \mathbf{h}])))). \quad (26b)$$

$$\mathcal{F}_i(\mathbf{x}) = \tanh(\mathbf{W}_i \mathbf{x} + \mathbf{b}_i) \quad (26c)$$

where $\mathbf{W}_i$ and $\mathbf{w}$ are matrices and vectors, respectively, without negative numbers, $\mathbf{b}_i$ and $b$ are biases, and $l$ is the number of non-negative fully-connected layers. With the two expressions above, we reveal the restrictions that FENN fails to compel:

1. The IEM must output 0 when the input is exactly $t_l$. Otherwise, the model allows future events to occur before the latest historical event, which is unreasonable. Unfortunately, as $\text{softplus}(x) = \log(1 + e^x)$ is always positive, $\Lambda^*(t_l)$ would be close to, but never be 0.

2. The model's output must be unbounded, in other words, $\lim_{t \to +\infty} \Lambda^*(t) = +\infty$ because the cumulative distribution function, $P^*(t) = \int_{t_l}^{t} \sum_{n \in M} p^*(n, \tau) d\tau = 1 - \exp(-\Lambda^*(t))$, must converge to 1 as $t \to +\infty$ if assuming the next event always happens. However, similar to FullyNN, because FENN's activation function between the fully-connected layers is $\tanh(x)$, whose value domain is $(-1, 1)$, the upper bound of $\Lambda^*(t)$ exists as shown in Equation (27), resulting in an unnormalized probability distribution.

$$\Lambda^*(t)_{max} = \sum_{m \in M} \text{softplus}(\mathbf{w}_1^\top \mathbf{1} + b_1) < +\infty \tag{27}$$

Moreover, these restrictions are parameter-independent, meaning that one must directly impose them into the model structure, making them more difficult to deal with. In conclusion, although FENN can provide the conditional joint PDF $p^*(m, t)$, its structure is still faulty, which could lead to inferior performance.

## C    EXPERIMENT SETTINGS

### C.1    REAL-WORLD DATASETS

We use the following six datasets to evaluate the performance of RM-NEP solutions.

- *BookOrder dataset*(BO)(Du et al., 2016) logs the frequent stock transactions from NYSE. Each event (i.e., transaction) belongs to one of the two event types[8]: buy or sell. The number of events is 400K, and the average sequence length is 3,319 for the training and evaluation set and 829 for the test set. This dataset is released under the Apache-2.0 license(Xue et al., 2023).

- *Retweet dataset*(Zhao et al., 2015) records when users Retweet a particular message on Twitter. This dataset distinguishes all users into three different types: (1) normal user, whose followers count is lower than the median, (2) influence user, whose followers count is higher than the median but lower than the 95th percentile, (3) famous user, whose followers count is higher than the 95th percentile. About 2 million Retweets are recorded, and the average sequence length is 108.This dataset is released under the Apache-2.0 license(Xue et al., 2023).

- *StackOverflow dataset*(SO)(Leskovec & Krevl, 2014) was collected from Stackoverflow[9], a popular question-answering website about various topics. Users providing decent answers will receive different badges as rewards. This dataset collects the timestamps when people obtain 22 badges from the website, and the average sequence length is 72.This dataset is released under the Apache-2.0 license(Xue et al., 2023).

- *Taobao*(Alibaba, 2018) records users' interactions on Taobao, an online shopping website from China. These actions include user clicking and buying online items, viewing reviews and comments, or searching for items. The average length of sequences in this dataset is 58, and 17 different marks are available. This dataset is released under the Apache-2.0 license(Xue et al., 2023).

- *USearthquake*(Xue et al., 2023) records all earthquakes happened in the continental US from USGS[10]. This dataset has 7 marks, referring to earthquakes with magnitude 2.0 to 2.9, 3.0 to 3.9, 4.0 to 4.9, 5.0 to 5.9, 6.0 to 6.9, 7.0 to 7.9, or 8 and higher. The average sequence length is 16.This dataset is released under the Apache-2.0 license(Xue et al., 2023).

- *Yelp*[11] records user comments mainly about restaurants in the United States. We categorize users into three groups: (1) users whose number of comments is lower

---

[8]In this study, event types and event marks are equivalent

[9]https://StackOverflow.com/

[10]http://earthquake.usgs.gov/earthquakes/eqarchives/year/eqstats.php

[11]https://www.Yelp.com/dataset

than the median (5 comments), (2) users whose number of comments is higher than the median but lower than the 95th percentile(92 comments), (3) users whose number of comments is higher than the 95th percentile. We pick stores which receive over 75 comments from Jan 1, 2020, and all comments of a store construct a sequence. This dataset includes 410,447 reviews involving 4,028 stores from 12 states. The average sequence length is 102. This dataset is released under Yelp's private license at `https://s3-media0.fl.yelpcdn.com/assets/srv0/engineering_pages/f64cb2d3efcc/assets/vendor/Dataset_User_Agreement.pdf`.

## C.2 SYNTHETIC DATASETS

All synthetic datasets are generated so we do not have any licenses information for them. The code to generate all synthetic datasets comes from the codebase of (Omi et al., 2019) at `https://github.com/omitakahiro/NeuralNetworkPointProcess` which is publicly accessible without any licenses.

- *Hawkes process dataset Hawkes_1* was generated utilising Hawkes process:

$$\lambda^*(t) = \mu_0 + \sum_{t_i < t} a \exp(-b(t - t_i)) \tag{28}$$

  where $\mu = 0.2$, $a = 0.8$, and $b = 1.0$.

- *Hawkes process dataset Hawkes_2* was generated utilising Hawkes process:

$$\lambda^*(t) = \mu_0 + \sum_{t_i < t} a_1 \exp(-b_1(t - t_i)) + a_2 \exp(-b_2(t - t_i)) \tag{29}$$

  where $\mu = 0.2$, $a_1 = a_2 = 0.4$, $b_1 = 1.0$, and $b_2 = 20$.

- *Homogeneous Poisson process dataset* was generated using the Homogeneous Poisson process where the conditional intensity function $\lambda^*(t)$ is constant over the entire timeline. This paper assumes $\lambda^*(t) = 1$.

- *Self-correct process dataset* was generated using the temporal point process whose intensity significantly drops when an event happens. The definition of the conditional intensity function is $\lambda^*(t) = \exp(\mu(t - t_i) - \alpha N)$ where $N$ is the number of occurred events, and $\mu$ and $\alpha$ are fixed parameters. In our experiments, we set $\alpha = \mu = 1$.

- *Stationary renewal process dataset* was generated using stationary renewal process, which directly defines the probability distribution over time $p^*(t)$ as a log-normal distribution as shown in Equation (30).

$$p^*(t|\sigma) = \frac{1}{\sigma t \sqrt{2\pi}} \exp(-\frac{\log^2(t)}{2\sigma^2}) \tag{30}$$

  where $\sigma$ is the standard deviation. Here, we set $\sigma = 1$. With Equation (30) and TPP's definition, one could solve the corresponding intensity function by Wolframalpha[12]:

$$\lambda^*(t) = \frac{-0.797885 \exp(-0.5 \log^2(t))}{-t + t \operatorname{erf}(0.707107 \log(t))} \tag{31}$$

  where $\operatorname{erf}(x) = \frac{2}{\sqrt{\pi}} \int_0^x \exp(-t^2) dt$.

These five synthetic distributions cooperate with a synthetic marking methods. This method generates discrete marks sampled from a uniform distribution. All synthetic datasets have 5 different marks.

## C.3 METRICS

### C.3.1 METRICS FOR SYNTHETIC DATASETS

For synthetic datasets, the real distribution $\hat{p}^*(m, t)$ is known. We can compare the generated $p^*(m, t)$ against the real one. Most papers report the relative NLL loss, that is, the average of the

---

[12]https://www.wolframalpha.com

absolute difference between $-\log \hat{p}^*(m,t)$ and $-\log p^*(m,t)$ on the observed events(if markers are unavailable, $-\log \hat{p}^*(t)$ and $-\log p^*(t)$(Omi et al., 2019; Shchur et al., 2020)). The lower relative NLL loss indicates a better performance. However, such a metric only evaluates performance at discrete events, which cannot gauge the overall discrepancy between $\hat{p}^*(m,t)$ and $p^*(m,t)$. So, this paper selects Spearman Coefficient $\rho$ and $L^1$ distance to measure the discrepancy between $\hat{p}^*(m,t)$ and $p^*(m,t)$ over time, while we also report the relative NLL loss for reference.

*Spearman Coefficient* $\rho(X,Y)$ measures the relationship between two arbitrary value sequences, $X$ and $Y$, as defined by Equation (32). If $X$ and $Y$ are more correlated, $\rho(X,Y)$ is higher; lower otherwise. Compared with the Pearson coefficient which is suitable if the relationship between $X$ and $Y$ is linear, Spearman coefficient could better deal with non-linear relationships. Because most probability distributions of TPP are non-linear, we select Spearman coefficient.

$$\rho(X,Y) = \frac{\text{Cov}(\text{Rank}(X), \text{Rank}(Y))}{\sigma_X \sigma_Y} \in [-1, 1] \tag{32}$$

where $\sigma_X$ and $\sigma_Y$ are the standard deviations of the values in sequence $X = \{x_1, x_2, \cdots, x_n\}$ and $Y = \{y_1, y_2, \cdots, y_n\}$, respectively. We expect $\rho$ between $\hat{p}^*(m,t)$ and $p^*(m,t)$ is close to 1.

$L^1$ *distance* measures how different two arbitrary functions are in interval $[a, b]$.

$$L^1(f,g) = \int_a^b |f(x) - g(x)| dx \geqslant 0 \tag{33}$$

The smaller the $L^1$ distance is, the more similar $f(x)$ and $g(x)$ are. When $L^1(f,g) = 0$, $f(x)$ almost equals to $g(x)$ in interval $[a, b]$ for any $f(x)$ and $g(x)$, or $f(x) = g(x)$ at every $x \in [a, b]$ if both $f(x)$ and $g(x)$ are continuous.

## C.4 BASELINES

The details of baselines in Section 4 are introduced next.

- *Fully Neural Network(FullyNN)*(Omi et al., 2019) uses a neural network to estimate the integral of $\lambda^*(t)$ for the history embedding $\mathbf{h}$ and inter-event time $t$. Then the density function is formulated to predict the time of the next event. FullyNN is designed for TPP without the information of event marks. To work with MTPP, FullyNN can be simply extended but it has a performance issue. Details are available in Appendix B.1. We rewrote FullyNN in PyTorch(Paszke et al., 2019) based on the official implementation available at `https://github.com/omitakahiro/NeuralNetworkPointProcess`, which is publicly accessible without any license.

- *Fully Event Neural Network(FENN)* is an extension of FullyNN. FENN successfully overcomes the computation graph overlap issue yet still inherits FullyNN's drawback of failing to comply with the mathematical restrictions. We believe sometimes such failure might be responsible for the inferior performance of FENN. Detailed information about FENN is available in Appendix B.2. The implementation of FENN is a direct modification of our FullyNN implementation.

- *Transformer Hawkes Process(THP)*(Zuo et al., 2020) uses a Transformer-based encoder to represent history as a hidden state $\mathbf{h}$. The softplus-based intensity function and the density function are modelled to predict the time of next event. We reproduce this model in PyTorch based on the paper.

- *Self-Attentive Hawkes Process(SAHP)*(Zhang et al., 2020) is based on the same intuition as Continuous-time LSTM(CTLSTM)(Mei & Eisner, 2017), which generalizes the classical Hawkes process by parameterizing its intensity function with recurrent neural networks. CTLSTM is an interpolated version of the standard LSTM, allowing us to generate outputs in a continuous-time domain. SAHP further improves performance by replacing LSTM with Transformers. Because the only difference between SAHP and CTLSTM is the history encoder, and SAHP has reported achieving better performance than CTLSTM, we only evaluate SAHP in this paper. We reproduce this model in PyTorch based on the paper.

- *Marked LogNormMix(Marked-LNM)*(Waghmare et al., 2022) is an MTPP extension of the LogNormMix(Shchur et al., 2020). Marked-LNM also follows the MT paradigm by

modeling $p^*(m)$ first, then using a composition of log Gaussian distribution to represent $p^*(t|m)$. To the best of our knowledge, Marked-LNM is the only MTPP approach predicting the mark of the next event first and then predicting the time of the event. However, Marked-LNM limits the form of $p^*(t|m)$ as the composition of log Gaussian distributions. This setting introduces inductive biases into the model, which could compromise the model prediction performance. We implement this model in PyTorch by modifying the official LogNormMix code at https://github.com/shchur/ifl-tpp. The official codes are released under the MIT license.

FENN, FullyNN, SAHP, and THP are CIF-based MTPP models. To solve RM-NEP with these models, we first estimate the value of $p^*(m)$ by the following equation:

$$
\begin{aligned}
p^*(m) &= \int_{t_l}^{+\infty} p^*(m,\tau)d\tau \\
&= \int_{t_l}^{+\infty} \lambda^*(m,\tau)\exp(-\int_{t_l}^{\tau} \lambda^*(m,\mathtt{t})d\mathtt{t})d\tau \\
&\approx \int_{t_l}^{T_{max}} \lambda^*(m,\tau)\exp(-\int_{t_l}^{\tau} \lambda^*(m,\mathtt{t})d\mathtt{t})d\tau
\end{aligned}
\tag{34}
$$

where $T_{max}$ is a predefined large value to approximate the infinity in the original expression of $p^*(m)$. We decide its value by the following equation.

$$
T_{max} = \min(10^6, \bar{t} + 10\sigma) \tag{35}
$$

where $\bar{t}$ and $\sigma$ are the mean and the standard deviation of the time intervals extracted from a dataset. This means every dataset has its $T_{max}$. As for the integration functions in Equation (34), we use the following equation to estimate them:

$$
\int_a^b f(x)dx \approx \sum_{i=0}^{N-1} f(a + i\frac{b-a}{N})\frac{b-a}{N} \tag{36}
$$

where $N$ means we equally divide the integral interval $[a,b]$ into $N$ sub-intervals. If increasing $N$, the estimation of $p^*(m)$ is more accurate but the computation cost and storage requirement increase quadruply, which quickly overwhelms an A100 GPU. Thus, we limit the value of $N$ for each mark $m$ while calculating $p^*(m)$ by the following equation:

$$
N_m = \min(T_{max} \times 200, \frac{C}{L}) \tag{37}
$$

Where $C = 3 \times 10^7$ controls the overall memory usage, $L$ is the length of the event sequence. So, if the event sequence is long, we use a smaller $N_m$ to reduce compute resource requirement and use a larger $N_m$ if the event sequence is short. It allows us to fully exploit the capacity of the GPU. Detailed information about the value of $N_m$ and $T_{max}$ are available in Table 5.

Table 5: The value of $N_m$ and $T_{max}$ of each real-world dataset used in experiments

|  | BO | Retweet | SO | Taobao | USeq | Yelp |
|---|---|---|---|---|---|---|
| Average of time interval $\bar{t}$ | 1.3273 | 2550.2 | 0.8167 | 4.1207 | 1.2187 | 7.2644 |
| Standard deviation of time interval $\sigma$ | 20.240 | 16230 | 1.0333 | 25.176 | 1.8454 | 13.410 |
| $T_{max}$ | 203.73 | 164847 | 11.150 | 255.88 | 19.673 | 144.37 |
| $N_m$(Average) | 18094 | 113048 | 2593 | 31748 | 3934 | 28273 |
| $N_m$(Maximum) | 18094 | 200000 | 2593 | 51175 | 3934 | 28273 |
| $N_m$(Minimum) | 18094 | 37878 | 2593 | 27573 | 3934 | 28273 |

After we obtain $p^*(m)$, we can calculate $F^*(t|m)$. According to the definition of MTPP and Equation (7), we have:

$$
\begin{aligned}
F^*(t|m) &= \frac{F^*(m,t)}{p^*(m)} = \frac{1}{p^*(m)}\int_{t_l}^t p^*(m,\tau)d\tau \\
&= \frac{1}{p^*(m)}\int_{t_l}^t p^*(m,\tau)d\tau \\
&= \frac{1}{p^*(m)}\int_{t_l}^t \lambda^*(m,\tau)\exp(-\int_{t_l}^{\tau} \lambda^*(m,\mathtt{t})d\mathtt{t})d\tau
\end{aligned}
\tag{38}
$$

Similar to $p^*(m)$, we estimate all integrals in Equation (38) by Equation (36) where the setting of $N$ is needed. Here, $N$ is denoted as $N_t$ to distinguish from the one in Equation (36).

$$N_t = \min(\min(\bar{t} \times 200, 500), \frac{C}{L * |\mathrm{M}| * |\mathrm{M}|}) \tag{39}$$

where $\bar{t}$ is the same one in Equation (35), $C$ and $L$ share the same meaning and value as those in Equation (37), and $|\mathrm{M}|$ is the number of marks in the dataset. Detailed information about $N_t$ is available in Table 6.

Table 6: The value of $N_t$ and $\bar{t}$ of each real-world dataset used in experiments

|  | BO | Retweet | SO | Taobao | USeq | Yelp |
|---|---|---|---|---|---|---|
| Average of time interval $\bar{t}$ | 1.3273 | 2550.2 | 0.8167 | 4.1207 | 1.2187 | 7.2644 |
| $N_t$(Average) | 265 | 500 | 174 | 500 | 243 | 500 |
| $N_t$(Maximum) | 265 | 500 | 174 | 500 | 243 | 500 |
| $N_t$(Minimum) | 265 | 500 | 174 | 500 | 243 | 500 |

With $F^*(t|m)$ finally available, we can solve Equation (6) for drawing samples from $p^*(t|m)$ to predict the time of the next event on the condition that its mark is $m$, as described in Section 3.1.

Different from CIF-based baselines discussed above, for each mark $m$, Marked-LNM(Waghmare et al., 2022) directly learns $p^*(m)$ by a classifier and $p^*(t|m)$ by LogNormMix. That is, Marked-LNM can give the value of $p^*(m)$ straightforwardly and direly draws samples from $p^*(t|m)$ to predict the time of the next event on the condition that its mark is $m$.

## C.5 DATA PREPROCESSING

We prepare synthetic and real-world datasets with normalization. For each dataset, normalization scales the time $t$ of every event in each event sequence by the time mean $\bar{t}$ of all events in all event sequences and standard deviation $\sigma$, as shown in Equation (40):

$$t_{scaled} = \frac{t - \bar{t}}{\sigma} \tag{40}$$

Normalization is useful when the time is relatively large, such as in the Retweet dataset. Table 7 shows how normalization is applied on various datasets.

Table 7: Data preprocessing.

| Dataset | BookOrder | Retweet | StackOverflow | Taobao | USearthquake | Yelp | five synthetic datasets |
|---|---|---|---|---|---|---|---|
| Normalization | ✓ | ✓ | ✓ | ✓ | ✓ | ✓ | ✗ |

Our work focuses on predicting when the next event will happen provided a mark, especially a rare mark. For each dataset, we classify if one mark is rare or frequent. The percentages of marks in each dataset are presented in Figure 5. Table 8 shows which marks are classified as frequent and which are classified as rare. BookOrder does not have rare/frequent marks because it has equally distributed marks.

Table 8: Rare marks and frequent marks.

| Dataset name | The number of marks | Rare Mark | Frequent Mark |
|---|---|---|---|
| BookOrder | 2 | [1] | [0] |
| Retweet | 3 | [2] | [0, 1] |
| StackOverflow | 22 | [1, 2, 6, 7, 9, 10, 11, 12, 13, 14, 15, 16, 17, 18, 19, 20, 21] | [0, 3, 4, 5, 8] |
| Taobao | 17 | [0, 1, 2, 3, 4, 5, 6, 7, 8, 9, 10, 11, 12, 13, 14, 15] | [16] |
| USearthquake | 7 | [3, 4, 5, 6] | [0, 1, 2] |
| Yelp | 3 | [0, 2] | [1] |

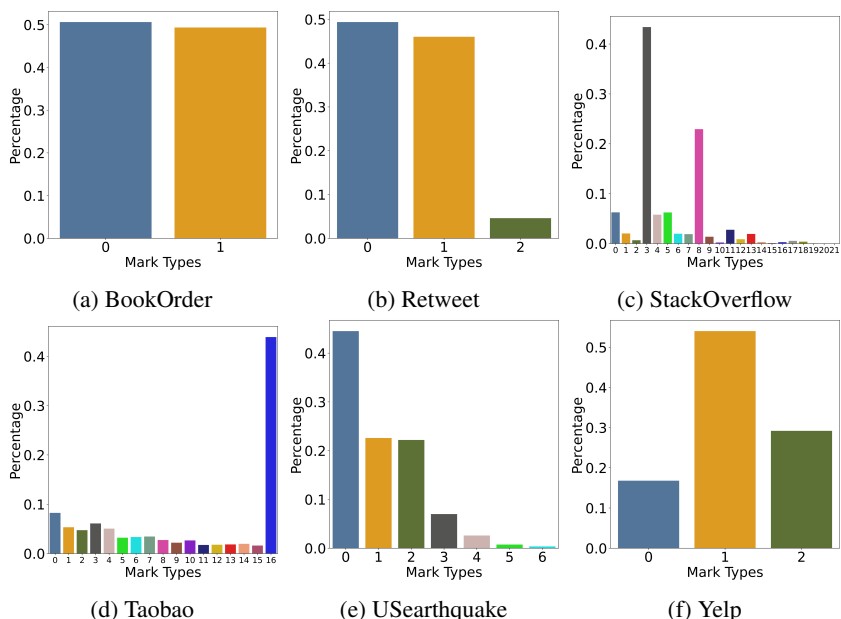

Figure 5: The frequency distribution of marks in real-world datasets.

### C.6 Model Training

This section introduces the hyperparameter settings for all MTPP models used in this paper. The two values of "Steps" refer to the number of warm-up steps and total training steps, respectively. "BS" refers to batch size, and "LR" refers to the learning rate. Unless otherwise specified, we repeatedly train a model 3 times with different random seeds and report the mean and standard deviation of the results. We conduct all experiments on an internal cluster. It includes Intel Xeon CPUs and NVIDIA A100-PCIE GPUs. All codes will be release upon acceptance under the MIT license.

For each mark $m$, we sample $N$ times $\{t^i\}_N^m$ from $F^*(t|m)$ to predict the time of the next event on the condition that its mark is $m$ by the inverse transform sampling:

$$F^*(t^i|m) = u^i \tag{41}$$

where $u^i$ is a random sample from a uniform distribution. The common practice samples $u_i$ from the standard uniform distribution $u^i \sim \mathcal{U}(0,1)$. MTPP allows $t_i$ to go to positive infinity. When $u^i$ is very close to 1, the time drawn from Equation (41) will be meaninglessly big and cause a negative impact to the accuracy of evaluation. To avoid this, we let $u^i \sim \mathcal{U}(0, 0.9)$. We find this trick can significantly stabilize the sampling process.

#### C.6.1 IFNMTPP Configurations

Table 9 lists the hyperparameter settings for IFNMTPP. The three values of "MS" (model structure) refer to the number of dimensions for history embedding $\mathbf{h}$, the number of dimensions for $\mathbf{v}_m$ and $\mathbf{b}_m$[13], and the number of non-negative fully-connected layers in the IEM module, respectively.

#### C.6.2 FullyNN and FENN Configurations

Table 10 shows hyperparameter settings for FullyNN and FENN. The three numbers in column "MS" share the same meaning as those in IFNMTPP.

#### C.6.3 THP Configurations

Table 11 shows all hyperparameter settings for THP. The six values of "MS" are the number of dimensions of the Transformer input vectors, the number of dimensions of the hidden outputs from

---

[13]$\mathbf{v}_m$ and $\mathbf{b}_m$ always have the same number of dimensions.

Table 9: Hyperparameter settings for IFNMTPP.

| Datasets | Steps | MS | BS | LR |
|---|---|---|---|---|
| Retweet | [80,000, 400,000] | [32, 16, 4] | 32 | 0.002 |
| Stackoverflow | [40,000, 200,000] | [32, 32, 2] | 32 | 0.002 |
| Taobao | [16,000, 80,000] | [32, 16, 4] | 32 | 0.002 |
| BookOrder | [4,000, 20,000] | [32, 32, 2] | 8 | 0.002 |
| Yelp | [40,000, 200,000] | [32, 16, 4] | 32 | 0.002 |
| USearthquake | [40,000, 200,000] | [32, 16, 4] | 32 | 0.002 |
| Synthetic | [20,000, 100,000] | [32, 64, 3] | 32 | 0.002 |

Table 10: Hyperparameter settings for FullyNN and FENN.

| Datasets | Steps | MS | BS | LR |
|---|---|---|---|---|
| Retweet | [80,000, 400,000] | [32, 16, 4] | 32 | 0.002 |
| Stackoverflow | [40,000, 200,000] | [32, 32, 2] | 32 | 0.002 |
| Taobao | [16,000, 80,000] | [32, 16, 4] | 32 | 0.002 |
| BookOrder | [4,000, 20,000] | [32, 32, 2] | 8 | 0.002 |
| Yelp | [40,000, 200,000] | [32, 16, 4] | 32 | 0.002 |
| USearthquake | [40,000, 200,000] | [32, 16, 4] | 32 | 0.002 |
| Synthetic | [20,000, 100,000] | [32, 64, 3] | 32 | 0.002 |

an RNN which is on top of the Transformer encoder, the number of dimensions of the vectors used by self-attentions ($q$, $k$, and $v$), the number of Transformer layers, and heads.

Table 11: Hyperparameter settings for THP.

| Datasets | Steps | MS | BS | LR |
|---|---|---|---|---|
| Retweet | [80,000, 400,000] | [16, 16, 32, 8, 3, 3] | 32 | 0.002 |
| Stackoverflow | [40,000, 200,000] | [16, 16, 32, 8, 3, 3] | 32 | 0.002 |
| Taobao | [16,000, 80,000] | [16, 16, 32, 8, 3, 3] | 32 | 0.002 |
| BookOrder | [4,000, 20,000] | [16, 16, 32, 8, 3, 4] | 8 | 0.002 |
| Yelp | [40,000, 200,000] | [16, 16, 32, 8, 3, 3] | 32 | 0.002 |
| USearthquake | [40,000, 200,000] | [16, 16, 32, 8, 3, 3] | 32 | 0.002 |
| Synthetic | [20,000, 100,000] | [16, 32, 64, 16, 3, 4] | 32 | 0.002 |

### C.6.4 SAHP CONFIGURATIONS

The hyperparameter settings for SAHP are available in Table 12. The first six values of "MS" share the same meaning as those in THP while the last is the dropout rate.

Table 12: Hyperparameter settings for SAHP.

| Datasets | Steps | MS | BS | LR |
|---|---|---|---|---|
| Retweet | [80,000, 400,000] | [16, 16, 32, 8, 3, 3, 0.1] | 32 | 0.002 |
| Stackoverflow | [40,000, 200,000] | [16, 16, 32, 8, 3, 3, 0.1] | 32 | 0.002 |
| Taobao | [16,000, 80,000] | [16, 16, 32, 8, 3, 3, 0.1] | 32 | 0.002 |
| BookOrder | [4,000, 20,000] | [16, 16, 32, 8, 3, 4, 0.1] | 8 | 0.002 |
| Yelp | [40,000, 200,000] | [16, 16, 32, 8, 3, 3, 0.1] | 32 | 0.002 |
| USearthquake | [40,000, 200,000] | [16, 16, 32, 8, 3, 3, 0.1] | 32 | 0.002 |
| Synthetic | [20,000, 100,000] | [16, 32, 64, 16, 3, 4, 0.1] | 32 | 0.002 |

### C.6.5 MARKED-LNM CONFIGURATIONS

The hyperparameter settings for Marked-LNM are presented in Table 13. The three values of "MS" are the number of the dimensions of LSTM, the number of the dimensions of mark embedding, and the number of Gaussian distributions, respectively.

Table 13: Hyperparameter settings for Marked-LNM.

| Datasets | Steps | MS | BS | LR |
|---|---|---|---|---|
| Retweet | [80,000, 400,000] | [32, 32, 16] | 32 | 0.002 |
| Stackoverflow | [40,000, 200,000] | [32, 32, 16] | 32 | 0.002 |
| Taobao | [16,000, 80,000] | [32, 32, 16] | 32 | 0.002 |
| BookOrder | [4,000, 20,000] | [32, 32, 16] | 8 | 0.002 |
| Yelp | [40,000, 200,000] | [32, 32, 16] | 32 | 0.002 |
| USearthquake | [40,000, 200,000] | [32, 32, 16] | 32 | 0.002 |
| Synthetic | [20,000, 100,000] | [32, 32, 16] | 32 | 0.002 |

# D ADDITIONAL EXPERIMENT RESULTS

## D.1 PERFORMANCE OF IFNMTPP FOR MODELING $p^*(m, t)$

IFNMTPP is designed to solve RM-NEP. For a better solution, the main purpose of IFNMTPP is to model the improper integration of $p^*(m, t)$. The advantage has been verified by the experiment results reported in Table 1. IFNMTPP models $p^*(m, t)$ at the same time while modeling the improper integration of $p^*(m, t)$. Compared to other existing MTPP models, the performance of IFNMTPP in modeling $p^*(m, t)$ is evaluated and reported in Table 14. The evaluation metric is NLL loss, the average of the $-\log p^*(m, t)$ at the observed events. The lower NLL loss indicates a better performance. We can observe that IFNMTPP shows a competent performance.

Table 14: Accuracy of $p^*(m, t)$ measured by NLL loss on real-world datasets. Lower is better.

| | IFNMTPP (Ours) | FENN | FullyNN | SAHP | THP | Marked-LNM |
|---|---|---|---|---|---|---|
| BookOrder | $-0.0963_{\pm 0.0151}$ | $-0.5504_{\pm 0.0521}$ | $-0.4819_{\pm 0.0195}$ | $-0.1580_{\pm 0.0689}$ | $4.4523_{\pm 0.8621}$ | $\mathbf{-1.8623_{\pm 0.0231}}$ |
| Retweet | $6.3225_{\pm 0.0007}$ | $6.3535_{\pm 0.0090}$ | $6.6437_{\pm 0.0380}$ | $\mathbf{6.1935_{\pm 0.0184}}$ | $10.379_{\pm 0.5349}$ | $6.5292_{\pm 0.0064}$ |
| Stackoverflow | $\mathbf{2.0540_{\pm 0.0029}}$ | $2.9126_{\pm 0.0078}$ | $3.6984_{\pm 0.0022}$ | $2.0713_{\pm 0.0028}$ | $2.5565_{\pm 0.0216}$ | $2.0992_{\pm 0.0014}$ |
| Taobao | $-0.7762_{\pm 0.0565}$ | $0.1644_{\pm 0.1989}$ | $-0.0431_{\pm 0.0484}$ | $\mathbf{-1.2779_{\pm 0.0421}}$ | $140.91_{\pm 81.166}$ | $1.2720_{\pm 0.1300}$ |
| USearthquake | $\mathbf{1.3278_{\pm 0.0533}}$ | $1.6582_{\pm 0.0363}$ | $1.8664_{\pm 0.0649}$ | $1.3544_{\pm 0.0300}$ | $2.0744_{\pm 0.3174}$ | $1.8514_{\pm 0.0462}$ |
| Yelp | $3.6542_{\pm 0.0003}$ | $3.7231_{\pm 0.0027}$ | $3.7912_{\pm 0.0024}$ | $3.6557_{\pm 0.0004}$ | $3.7036_{\pm 0.0144}$ | $\mathbf{3.6363_{\pm 0.0040}}$ |

## D.2 TIME PREDICTIONS BY $p^*(t|m)$ V.S. TIME PREDICTIONS BY $p^*(t)$

Is the time prediction by sampling $p^*(t|m)$ more accurate than that by sampling $p^*(t)$? Specifically, one might come up with an intuitive formulation of RM-NEP that treats the predicted time by $p^*(t)$ as the predicted time for all marks. We compare this intuitive formulation with ours by $\text{MMAE}_\text{M}$ on six real-world datasets. To be fair, both $p^*(t|m)$ and $p^*(t)$ are produced using IFNMTPP. The results are in the Table 15. We can observe that sampling from $p^*(t|m)$ provides a generally more accurate time prediction when solving RM-NEP.

Table 15: Time prediction by sampling $p^*(t)$ v.s. that by sampling $p^*(t|m)$ on real-world datasets, measured by MMAE, lower is better, $p^*(t|m)$ and $p^*(t)$ are produced using IFNMTPP.

| | | BO | Retweet | SO | Taobao | USearthquake | Yelp |
|---|---|---|---|---|---|---|---|
| Time Prediction by $p^*(t|m)$ | $\text{MMAE}_\text{M}$ | $\mathbf{1.1967_{\pm 0.0076}}$ | $2515.1_{\pm 6.5029}$ | $\mathbf{0.5212_{\pm 0.0142}}$ | $0.3324_{\pm 0.0579}$ | $\mathbf{0.6856_{\pm 0.0063}}$ | $\mathbf{5.3271_{\pm 0.0066}}$ |
| | $\text{MMAE}_{\text{M}_r}$ | $1.1211_{\pm 0.0120}$ | $3291.2_{\pm 29.097}$ | $\mathbf{0.4986_{\pm 0.0175}}$ | $0.3385_{\pm 0.0627}$ | $\mathbf{0.6966_{\pm 0.0081}}$ | $5.3888_{\pm 0.0078}$ |
| | $\text{MMAE}_{\text{M}_f}$ | $\mathbf{1.2775_{\pm 0.0093}}$ | $\mathbf{2198.7_{\pm 2.4798}}$ | $\mathbf{0.6063_{\pm 0.0001}}$ | $0.2529_{\pm 0.0055}$ | $\mathbf{0.6713_{\pm 0.0048}}$ | $\mathbf{5.2059_{\pm 0.0071}}$ |
| Time Prediction by $p^*(t)$ | $\text{MMAE}_\text{M}$ | $1.2011_{\pm 0.0064}$ | $\mathbf{2496.5_{\pm 2.7826}}$ | $0.6417_{\pm 0.0127}$ | $\mathbf{0.2420_{\pm 0.0227}}$ | $0.8516_{\pm 0.2378}$ | $5.3314_{\pm 0.0135}$ |
| | $\text{MMAE}_{\text{M}_r}$ | $\mathbf{1.1189_{\pm 0.0094}}$ | $3215.7_{\pm 4.3992}$ | $0.6515_{\pm 0.0167}$ | $\mathbf{0.2411_{\pm 0.0215}}$ | $0.7008_{\pm 0.0075}$ | $\mathbf{5.3670_{\pm 0.0097}}$ |
| | $\text{MMAE}_{\text{M}_f}$ | $1.2893_{\pm 0.0050}$ | $2199.7_{\pm 3.3627}$ | $0.6095_{\pm 0.0008}$ | $\mathbf{0.2573_{\pm 0.0429}}$ | $1.2295_{\pm 0.7947}$ | $5.2155_{\pm 0.0031}$ |

