# OpenReview forum: "Rare-Mark-Aware Next Event Prediction In Marked Event Streams"
_ICLR.cc/2025/Conference — Submitted to ICLR 2025_

### Official Review · Reviewer_oDxB · 2024-11-01

**Soundness:** 3
**Presentation:** 4
**Contribution:** 4
**Rating:** 8
**Confidence:** 4

**Summary:**

This paper focuses on utilizing Marked Temporal Point Process (MTPP) models to address the Next-event Prediction (NEP) problem. It highlights a primary challenge of NEP: the imbalanced distribution of mark types. To address this, the paper introduces a new problem, Rare-mark-aware Next Event Prediction (RM-NEP), which is designed to ensure that rare marks consistently appear in prediction results. The paper also presents a novel IFNMTPP model to resolve issues related to inadequate integration over infinite time intervals when estimating the probability of marks and their timing in RM-NEP.

**Strengths:**

1. The proposed RM-NEP problem offers fresh insights into the NEP challenge and the field of MTPP, presenting a potentially effective solution for addressing the issue of imbalanced mark types.
2. The paper is well-written, with a clear and fluent presentation of the NEP problem, its challenges, and the proposed solution.
3. The IFNMTPP model is straightforward in its design, with empirical studies demonstrating its superior efficiency.

**Weaknesses:**

1. Although the primary focus of the paper is on accurately predicting rare mark types, Table 3 suggests that IFNMTPP does not show significant superiority in mark prediction performance. Instead, its strengths appear more pronounced in time prediction and efficiency. The paper could benefit from more detailed experimental analysis regarding the accuracy of predicting rare mark types.
2. The illustration depicting the architecture of IFNMTPP could be refined to provide a clearer demonstration of its design.

**Questions:**

1. (Related to W1) Could the authors elaborate on how their experimental results empirically demonstrate the effectiveness of the proposed RM-NEP problem and IFNMTPP model in addressing the issue of missing rare marks in NEP?

---

> ### Author Response · Authors · 2024-11-21
> **Rebuttal to reviewer oDxB**
>
> We greatly appreciate your detailed and insightful review of our paper.
>
> > "Although the primary focus of the paper is on accurately predicting rare mark types, Table 3 suggests that IFNMTPP does not show significant superiority in mark prediction performance. Instead, its strengths appear more pronounced in time prediction and efficiency. The paper could benefit from more detailed experimental analysis regarding the accuracy of predicting rare mark types."
>
> It is inaccurate to say that "the primary focus of the paper is on accurately predicting rare mark types". We would like to stress that the primary focus of this paper is to solve RM-NEP. Specially, RM-NEP returns $p^*(m)$ and $\bar{t}_m$ for every mark $m$. Compared with baselines, IFNMTPP shows remarkable superiority in time prediction (Table 1) while a slight improvement in mark prediction (Table 3). Based on our analysis, the reason is that the mark prediction is less sensitive to the accuracy of $\Gamma^*(m, t)$ as time prediction. The detailed explanation is below.
>
> Our method is integral-free by using the proposed IFNMTPP. Unlike baselines, IFNMTPP avoids using numerical methods to solve $\Gamma^*(m, t)= \int_{t}^{+\infty}{p^*(m, \tau)d\tau}$. As a result, IFNMTPP provides an accurate $\Gamma^*(m, t)$ compared with baselines. Based on $\Gamma^*(m, t)$, we derive and report $p^*(m)$ (the probability that the mark of the next event is $m$) and $\bar{t}_m$ (the time of the next event if the mark is $m$) for each mark $m$.
>
> Table 1 reports the prediction accuracy of $\bar{t}_m$, which is the average of samples from $p^*(t|m)$. The samples drawn from $p^*(t|m)$ are based on the values of $\Gamma^*(m, t)$ at many different times while solving Equation (6) using bisection method. The accurate $\Gamma^*(m, t)$ will lead to the accurate $\bar{t}_m$. This is why the advantage of our method against baselines is remarkable as shown in Table 1.
>
> For calculating macro-F1 in Table 3, the mark with the highest $p^*(m)$ is selected as the mark prediction. $p^*(m)$ is the value of $\Gamma^*(m, t)$ at a single time $t=t_l$. More accurate $\Gamma^*(m, t)$ should lead to more accurate $p^*(m)$, which can help correctly predict the mark. However, the accuracy improvement on $p^*(m)$ is limited compared with that on $\bar{t}_m$. The reason is the mark prediction only involves the value of $\Gamma^*(m, t)$ at a single time $t=t_l$, while $\bar{t}_m$ is based on the value of $\Gamma^*(m, t)$ at many different times.
>
> > "The illustration depicting the architecture of IFNMTPP could be refined to provide a clearer demonstration of its design."
>
> Thanks! As advised, we have updated Figure 2 in the revised version with more details.
>
> > "(Related to W1) Could the authors elaborate on how their experimental results empirically demonstrate the effectiveness of the proposed RM-NEP problem and IFNMTPP model in addressing the issue of missing rare marks in NEP?"
>
> Because NEP only predicts a single mark and a single time, the frequent marks overwhelm the rare marks in the output of NEP. This is known as the rare mark missing issue. One typical way to solve the problem is to handle data imbalance (e.g., by oversampling or undersampling) to increase the chance of rare events in the output. In contrast, our RM-NEP problem adopts a different way to address the problem. Specifically, RM-NEP outputs $p^*(m)$ (the probability of the next event being $m$) and $\bar{t}_m$ (time the next event will happen provided its mark is $m$) for every mark $m$. Because every mark is already in the output, we do not need to increase the chance of rare events in the output.
>
> By outputting $p^*(m)$ and $\bar{t}_m$ for every mark $m$, RM-NEP addresses the rare mark missing issue in NEP.  Our focus is on increasing the accuracy of $p^*(m)$ and $\bar{t}_m$. To this end, IFNMTPP is developed to produce accurate $\Gamma^*(m, t)$ based on which $p^*(m)$ and $\bar{t}_m$ are derived. Compared with baselines, IFNMTPP-based solution shows remarkable superiority in time prediction (Table 1) and a slight improvement in mark prediction (Table 3). Based on our analysis, the reason is that the mark prediction is less sensitive to the accuracy of $\Gamma^*(m, t)$ as the time prediction. The detailed explanation can be found in our response to the reviewer's first comment.

---

> > ### Comment · Reviewer_oDxB · 2024-11-24
> >
> > I appreciate the authors' thorough efforts in providing additional analysis and information about their proposed method. The clarifications have addressed my concerns satisfactorily, and consequently, I have modestly increased my evaluation score.

---

### Official Review · Reviewer_o69X · 2024-11-01

**Soundness:** 3
**Presentation:** 3
**Contribution:** 2
**Rating:** 5
**Confidence:** 5

**Summary:**

This paper investigates how to reduce the problem of rare mark missing when event prediction is imbalanced, thereby reducing the risk of missing key events. The paper provides a detailed description of the proposed IFNMTPP method and conducts comparative experiments on multiple datasets. Results show the performance of IFNMTPP.

**Strengths:**

S1: The paper studies the RM-NEP problem, unifies abnormal integrals, and proposes IFNMTPP to ensure that the prediction results of rare marks are not missed when the marks are imbalanced.
S2. The paper is well-articulated, offering a clear explanation of the concepts and methodologies employed.
S3. Extensive experiments on real-world and synthetic datasets demonstrate the effectiveness of the proposed method.

**Weaknesses:**

W1. The purpose of this article is to improve the prediction accuracy of rare events. According to the experimental results of macro-F1 in Table 3, there is a slight improvement in the prediction accuracy of rare marks. In addition, earthquakes are unlikely to be accurately predicted through event prediction. Both the accuracy of frequent marks and rare marks before and after improvement are very low. Does this study have practical application value?
W2. Figure 2 is not very clear. It is recommended to refine it. The symbols inside are not consistent with the description in the text, such as v, s, and f.
W3. Incorrect punctuation is used in line 20 and line 78.

**Questions:**

Q1: BookOrder's mark type [1] account for over 40%. Does this meet the definition of the rare mark?

---

> ### Author Response · Authors · 2024-11-21
> **Rebuttal to reviewer o69X**
>
> We greatly appreciate your detailed and insightful review of our paper.
>
> > "W1. The purpose of this article is to improve the prediction accuracy of rare events. According to the experimental results of macro-F1 in Table 3, there is a slight improvement in the prediction accuracy of rare marks. In addition, earthquakes are unlikely to be accurately predicted through event prediction. Both the accuracy of frequent marks and rare marks before and after improvement are very low. Does this study have practical application value?"
>
> Our method is integral-free by using the proposed IFNMTPP. Unlike baselines, IFNMTPP avoids using numerical methods to solve $\Gamma^*(m, t)= \int_{t}^{+\infty}{p^*(m, \tau)d\tau}$. As a result, IFNMTPP provides an accurate $\Gamma^*(m, t)$ compared with baselines. Based on $\Gamma^*(m, t)$, we derive $p^*(m)$ directly and $\bar{t}_m$ by sampling for our method and baselines. Finally, we report $p^*(m)$ (the probability that the mark of the next event is $m$) and $\bar{t}_m$ (the time of the next event if the mark is $m$) for each mark $m$.
>
> For calculating macro-F1 in Table 3, the mark with the highest $p^*(m)$ is selected as the mark prediction. $p^*(m)$ is the value of $\Gamma^*(m, t)$ at a single time $t=t_l$. More accurate $\Gamma^*(m, t)$ should lead to more accurate $p^*(m)$, which can help correctly predict the mark. However, the accuracy improvement on $p^*(m)$ is limited compared with that on $\bar{t}_m$. The reason is the mark prediction only involves the value of $\Gamma^*(m, t)$ at a single time $t=t_l$, while $\bar{t}_m$ is based on the value of $\Gamma^*(m, t)$ at many different times.
>
> Although the marco-F1 values are low for the dataset Earthquakes, the practical value of this study remains the same. The earthquakes have 7 marks. The low prediction accuracy on frequent but minor earthquakes is less concerning. The low prediction accuracy on rare but major earthquakes is a big problem. In this situation, the next event prediction like NEP is risky, i.e., a single mark and a single time are returned as the prediction. Instead, the practical solution is to list $p^*(m)$ and $\bar{t}_m$ for every mark like our RM-NEP.
>
> > "W2. Figure 2 is not very clear. It is recommended to refine it. The symbols inside are not consistent with the description in the text, such as v, s, and f."
>
> Thanks for pointing this out. In the revised version, we have updated the Figure 2.
>
> > "W3. Incorrect punctuation is used in line 20 and line 78."
>
> Thanks for pointing this out. In the revised version, we have fixed them.
>
> > "Q1: BookOrder's mark type [1] account for over 40\%. Does this meet the definition of the rare mark?"
>
> There is not a definite threshold for imbalance across different problems. For comprehensive evaluation of our proposed method across datasets of varying levels of imbalance,  we purposedly include the BookOrder dataset with very low level of imbalance to show that our approach is robust  to datasets with low level of imbalance.

---

> > ### Comment · Reviewer_o69X · 2024-11-26
> >
> > Thanks for the response! However, the performance improvement is very limited. Thus, I would keep the score.

---

> > > ### Author Response · Authors · 2024-11-27
> > > **Response to the comment of Reviewer o69X**
> > >
> > > Many thanks for your feedback! There’s just one more thing we’d like to clarify. Our RM-NEP returns $p^*(m)$ (the probability that the mark of the next event is $m$) and $\bar{t}_m$ (the time of the next event if the mark is $m$) for every mark $m$. To evaluate the accuracy of $p^*(m)$, the mark with the highest $p^*(m)$ is used as the mark prediction like NEP. It is reasonable to evaluate in this way. Assuming we have an ideal model that can exactly estimate $p^*(m)$ for every mark $m$, the mark prediction accuracy will be 100%. Even though we don't have such an ideal model in practice, we expect the more accurate $p^*(m)$ to lead to a higher mark prediction accuracy.
> > >
> > > The absolute low accuracy in Table 3 highlights the challenging nature of mark prediction, particularly for datasets with more marks. As shown in Table 3, the mark prediction accuracy is much lower for datasets with more marks like StackOverflow, Taobao, and USearthquake. On these datasets, our method demonstrates more advantages compared with baselines. Considering the best and the second best performance on these datasets, our method wins 7, FENN wins 5, FullyNN wins 1, SAHP wins 3, THP wins 0, Marked-LNM wins 3. This is attributed to the better $p^*(m)$ estimation of our method.
> > >
> > > The absolute low accuracy in Table 3 also implies that mark prediction, i.e., returning the mark with the highest $p^*(m)$, is often less useful in practice. In contrast, returning $p^*(m)$ and $\bar{t}_m$ for every mark $m$ like our RM-NEP can provide more information to users. Finally, we would like to stress our method demonstrated significant superiority in time predication as in Table 1, our method is much faster than all baselines as in Table 2, and our method enjoys much higher model fidelity on synthetic datasets drawn from known distribution $p^*(m, t)$ as in Table 4.

---

### Official Review · Reviewer_ZLCK · 2024-11-03

**Soundness:** 3
**Presentation:** 3
**Contribution:** 3
**Rating:** 6
**Confidence:** 4

**Summary:**

This paper makes a substantial contribution to the field of MTPPs by addressing the rare mark missing issue and providing a computationally efficient solution through IFNMTPP. The work is theoretically robust, empirically validated, and has practical significance in domains where rare events play a critical role.

**Strengths:**

1.The paper provides a thorough theoretical foundation for the RM-NEP problem, including detailed derivations of the probability distributions and the integral-free approximation.

2.This paper proposes a novel approach, IFNMTPP, which avoids the computational burden of traditional numerical integration methods (e.g., Monte Carlo integration) by directly approximating the integral using a neural network. This is a computationally efficient solution that enables the model to handle large-scale datasets.

3.The authors conduct extensive experiments on various datasets, showing that their approach consistently outperforms existing baselines. The empirical results are strong and demonstrate the practical utility of the proposed method.

**Weaknesses:**

W1: If I understand correctly, RM-NEP assumes that rare marks can be predicted accurately by decoupling time and mark prediction. Does this assumption hold across different types of datasets, especially when the marks exhibit temporal correlations?

W2: The IFNMTPP model approximates improper integrals using a "monotonically decreasing neural network." However, the paper does not provide sufficient details about how this approximation is performed, nor does it explain the intuition behind why a monotonically decreasing function is appropriate.

**Questions:**

Q1: How interpretable are the results of RM-NEP, particularly for rare marks? Does the neural network-based approximation provide any insight into why a rare mark might be predicted?

Q2: The paper focuses on marked temporal point processes where marks are categorical. How well does the proposed method generalize to cases where the marks are continuous.

---

> ### Author Response · Authors · 2024-11-21
> **Rebuttal to reviewer ZLCK**
>
> We greatly appreciate your detailed and insightful review of our paper.
>
> > "W1: If I understand correctly, RM-NEP assumes that rare marks can be predicted accurately by decoupling time and mark prediction. Does this assumption hold across different types of datasets, especially when the marks exhibit temporal correlations?"
>
> If our understanding is correct, "decoupling time and mark prediction" means predicting the mark of the next event and predicting the time of the next event independently. If so, RM-NEP does not have the assumption. Precisely, RM-NEP models $\Gamma^*(m, t)= \int_{t}^{+\infty}{p^*(m, \tau)d\tau}$ using the proposed IFNMTPP to capture the pattern of mark and time simultaneously. Based on $\Gamma^*(m, t)$, RM-NEP obtains $p^*(m) = \int_{t_l}^{+\infty}{p^*(m, \tau)d\tau}$ (the probability that the mark of the next event is $m$) directly and $\bar{t}_m$ (the time of the next event if the mark is $m$) by sampling.
>
> > "W2: The IFNMTPP model approximates improper integrals using a "monotonically decreasing neural network." However, the paper does not provide sufficient details about how this approximation is performed, nor does it explain the intuition behind why a monotonically decreasing function is appropriate."
>
> In the revised version, to clarify how IFNMTPP approximates $\Gamma^*(m, t)$, we have updated Figure 2 by including more structural details of IFNMTPP, and have added more explanations in the section "Integral-Free Neural Marked Temporal Point Process (IFNMTPP)".
>
> First, we explain why a monotonically decreasing neural network is necessary. IFNMTPP approximates $\Gamma^*(m, t)=\int_{t}^{+\infty}{p^*(m, \tau)d\tau}$. The derivative of $\Gamma^*(m, t)$ w.r.t. $t$ is $-p^*(m, t)$. Probability density $p^*(m, t)$ is always positive. This means $\Gamma^*(m, t)$ is monotonically decreasing w.r.t. $t$. As the approximation network, IFNMTPP must be aligned with $\Gamma^*(m, t)$ to decrease monotonically.
>
> Next, we show how IFNMTPP is implemented as a monotonically decreasing network. The parameters in $\mathbf{v}_m$ and IEM are all non-negative. These settings ensure that network before $\sigma(x) = 1/(1 + e^x)$ has positive derivative w.r.t. $t$. Because $\sigma(x)$ is monotonically decreasing, we know by the chain rule that one $\sigma(x)$ is sufficient to flip the derivative from positive to negative, creating a monotonically decreasing model.
>
> > "Q1: How interpretable are the results of RM-NEP, particularly for rare marks? Does the neural network-based approximation provide any insight into why a rare mark might be predicted?"
>
> RM-NEP outputs $p^*(m)$ (the probability of the next event being $m$) and $\bar{t}_m$ (time the next event will happen provided its mark is $m$) for every mark $m$. Because every mark is already in the output, we do not need to increase the chance of rare events in the output.
>
> By outputting $p^*(m)$ and $\bar{t}_m$ for every mark $m$, RM-NEP addresses the rare mark missing issue in NEP.  Our focus is on increasing the accuracy of $p^*(m)$ and $\bar{t}_m$. To this end, IFNMTPP is developed to produce accurate $\Gamma^*(m, t)$ based on which $p^*(m)$ and $\bar{t}_m$ are derived. Compared with baselines, IFNMTPP-based solution shows remarkable superiority in time prediction (Table 1) and a slight improvement in mark prediction (Table 3). Based on our analysis, the reason is that the mark prediction is less sensitive to the accuracy of $\Gamma^*(m, t)$ as the time prediction.
>
> > "Q2: The paper focuses on marked temporal point processes where marks are categorical. How well does the proposed method generalize to cases where the marks are continuous."
>
> It is possible to generalize the proposed method where the marks are categorical to the cases where the marks are continuous. However, it is not straightforward, and a thorough study is needed.

---

> > ### Comment · Reviewer_ZLCK · 2024-11-23
> >
> > Thank you to the authors for their responses and for clarifying the questions regarding the methodology.
> >
> > The proposed approach is well-motivated to address RM-NEP. However, the improvement in mark prediction, as shown in Table 3, appears marginal. Additionally, IFNMTPP underperforms compared to baselines across all metrics on the Taobao and Yelp datasets in time prediction. Moreover, the standard deviation of IFNMTPP's accuracy in predicting rare marks is twice that of FENN on the US Earthquake dataset, demonstrating relatively poor stability of the IFNMTPP on this dataset. Due to these concerns regarding the accuracy and stability of IFNMTPP, I lower my score to 6. If the authors can provide more detailed analyses and practical solutions to demonstrate the model's superiority, I would be willing to reconsider and raise my score.

---

> > > ### Author Response · Authors · 2024-11-24
> > > **Response to Reviewer ZLCK's comment**
> > >
> > > Thanks for the insightful feedback.
> > >
> > > For mark prediction, we list the standard deviation for IFNMTPP and FENN in Table A below, where the lower standard deviation is highlighted. Across all 18 comparisons,  IFNMTPP wins on 10 and FENN wins on 8. Overall, it shows that IFNMTPP is comparably as stable as FENN.
> > >
> > > For time prediction, we conducted many runs of experiments. On Taobao, our IFNMTPP is reliably the second best. On Yelp, all methods including our IFNMTPP have comparable performance. In summary on all 6 datasets, our IFNMTPP wins on 4, FENN on 0, FullyNN on 1, SAHP on 0, THP on 0, and Marked-LNM on 1. We also note the "No Free Lunch" Theorem, where it is widely recognised that not a single machine learning algorithm can have the best performance across all problems. We believe the fact that IFNMTPP wins on 4 out of 6 datasets shows its superiority.
> > >
> > > Table A: This table compares IFNMTPP and FENN on standard deviations of mark prediction accuracy. Lower is better. The bold indicates the best values.
> > >
> > > | Model    | Metric      | BO              | Retweet          | SO               | Taobao             | USearthquake     | Yelp             |
> > > |---------------|-------------|-----------------|------------------|------------------|--------------------|------------------|------------------|
> > > | **Model (Ours)** | All Marks  | 0.6003±**0.0009** | 0.3569±**0.0001**  | 0.1519±0.0033    | 0.2338±0.0258      | 0.1795±**0.0078**  | 0.2524±**0.0009**  |
> > > |               | Rare Marks  | 0.7235±**0.0032** | 0.0014±0.0002    | 0.1457±0.0076    | 0.1324±**0.0054**    | 0.0012±0.0008    | 0.0376±**0.0019**  |
> > > |               | Frequent Marks | 0.7573±0.0043  | 0.5057±**0.0004**  | 0.1364±0.0009    | 0.3186±**0.0541**    | 0.2525±0.0110    | 0.7986±**0.0107**  |
> > > | **FENN**      | All Marks   | 0.3923±0.0580   | 0.3673±0.0007    | 0.0938±**0.0002**  | 0.1283±**0.0104**    | 0.1835±0.0079    | 0.2436±0.0029    |
> > > |               | Rare Marks  | 0.0408±0.0062   | 0.0013±**0.0000**  | 0.0298±**0.0015**  | 0.0210±0.0129      | 0.0006±**0.0004**  | 0.0160±0.0066    |
> > > |               | Frequent Marks | 0.9885±**0.0031** | 0.5195±0.0015    | 0.1512±**0.0003**  | 0.4252±0.2559      | 0.2587±**0.0101**  | 0.8540±0.0754    |

---

> > > ### Author Response · Authors · 2024-11-25
> > > **Response to Reviewer ZLCK's comment about table 3.**
> > >
> > > Here, we explain why mark prediction improvement using our solution is marginal compared with baselines, but time prediction improvement is remarkable. Our RM-NEP returns $p^*(m)$ and $\bar{t}_m$ for every mark $m$. To improve the accuracy of $p^*(m)$  (the probability that the mark of the next event is $m$) and $\bar{t}_m$ (the time of the next event if the mark is $m$) for every mark $m$, IFNMTPP is proposed. Unlike baselines, IFNMTPP avoids using numerical methods to solve . As a result, IFNMTPP provides an accurate $\Gamma^*(m, t) = \int_t^{+\infty}{p^*(m, \tau)d\tau}$ compared with baselines. Based on $\Gamma^*(m, t)$, we derive and report $p^*(m)$ and $\bar{t}_m$ for each mark $m$.
> > >
> > > Our RM-NEP returns $p^*(m)$ and $\bar{t}_m$ for every mark $m$. To evaluate the accuracy of $p^*(m)$ and $\bar{t}_m$ for every mark $m$, the mark with the highest $p^*(m)$ and its time are used as the mark prediction and time prediction of the next event like NEP. The time prediction accuracy is reported in Table 1 and the mark prediction accuracy in Table 3.
> > >
> > > In Table 1, the predicted time $\bar{t}_m$ for every mark $m$ is the average of samples drawn from $p^*(t|m)$. Specifically, the samples drawn from $p^*(t|m)$ are based on the values of $\Gamma^*(m, t)$ at many different times by solving Equation (6) with the bisection method. The accurate $\Gamma^*(m, t)$ will lead to the accurate $\bar{t}_m$. This is why the advantage of our method against baselines is remarkable as in Table 1.
> > >
> > > In Table 3, if mark $m$ has the highest $p^*(m)$ among all marks, $m$ is selected as the mark prediction. $p^*(m)$ is the value of $\Gamma^*(m, t)$ at a single time $t=t_l$. More accurate $\Gamma^*(m, t)$ should lead to more accurate $p^*(m)$, which can help correctly predict the mark. However, the accuracy improvement on $p^*(m)$ is limited compared with that on $\bar{t}_m$. The reason is the mark prediction only involves the value of $\Gamma^*(m, t)$ at a single time $t=t_l$, while $\bar{t}_m$ is based on the value of $\Gamma^*(m, t)$ at many different times.
> > >
> > > The marginal accuracy improvement on mark prediction shown in Table 3 verifies our belief that completely solving the rare mark missing issue in NEP is, if not impossible, highly challenging. In this situation, it is sensible to return $p^*(m)$ and $\bar{t}_m$ for every mark $m$ like our RM-NEP. Finally, we would like to stress running our method is much faster than all baselines as in Table 2.

---

> > > > ### Comment · Reviewer_ZLCK · 2024-11-25
> > > >
> > > > Thanks for the authors' response. Although the improvement in mark prediction is marginal, considering the contributions of this work to addressing RM-NEP, I would like to support the acceptance of this paper.

---

### Official Review · Reviewer_i8f9 · 2024-11-04

**Soundness:** 2
**Presentation:** 1
**Contribution:** 2
**Rating:** 3
**Confidence:** 4

**Summary:**

This paper solves a problem in marked event prediction when the distribution of marks is significantly imbalanced i.e., some marks are frequent, and others are rare. The paper introduces a problem namely Rare-mark-aware Next Event Prediction (RM-NEP) and solves the problem to answer two questions: “what is the probability that the mark of the next event is m? and if m, when will the next event happen?”. Solving RM-NEP gives rare marks equal opportunity as frequent marks in the next event prediction. This guarantees that rare marks are always included in the predicted results. To solve RM-NEP effectively, the authors first unify the improper integration of two different functions into one and then develop a novel Integral-free Neural Marked Temporal Point Process (IFNMTPP) to approximate the target integral directly.

**Strengths:**

1.	The problem is interesting.
2.	The Figures are intuitive.

**Weaknesses:**

1.	The main difference between the problem of the paper and the existing problem is not clear. For example, what are the differences between RM-NEP and rare event forecasting?
2.	The motivation of RM-NEP is not convincing. (i) If a mark is rare (i.e., it occurs very few times in the history). Then, it can be dominated by frequent marks in the prediction. This phenomenon is completely normal.  (ii) If a mark is rare and important compared to other marks, why don’t we only consider that mark as a single variable so that there is no imbalance anymore?
3.	The paper is not self-contained. For example, how the existing studies solve NEP is not clear. The authors only list a large number of papers in the Related Work section. Similarly, how the existing studies model MTPP is not clear. The authors only list a large number of papers in the Introduction section. A summarization and comparison are needed to provide a better understanding.
4.	Some words are hard to understand. For example, RMTPP is not defined.
5.	Some notations are not defined. For example, what is $\tau$?
6.	Intuitively, can we solve the problem by undersampling dominating marks?
7.	I cannot understand lines 297-299. If t=t_l then the integration equals 0.
8.	The main idea of using integral-free comes from FullyNN by using IEM. Basically, the authors adapt it to marked events, which is straightforward.
9.	The authors do not prove why using IEM can achieve the integral-free solution.
10.	There is no ablation study. For example, what is the performance of TFNMTPP with different imbance ratios?

**Questions:**

1.	If a mark is rare and important compared to other marks, why don’t we only consider that mark as a single variable such that there is no imbalancing anymore.
2.	What is the performance of TFNMTPP with different imbance ratios.
3.	Intuitively, can we solve the problem by undersampling dominating marks?

---

> ### Author Response · Authors · 2024-11-23
> **Rebuttal to Reviewer i8f9 (1/4)**
>
> We greatly appreciate your detailed and insightful review of our paper.
>
> > "W1: The main difference between the problem of the paper and the existing problem is not clear. For example, what are the differences between RM-NEP and rare event forecasting?"
>
> Rare event forecasting solves the problem that events with much fewer samples are dominated by events with more samples in the output of some machine learning tasks, e.g., classification [1]. Rare event forecasting overcomes data imbalance typically by undersampling or oversampling to increase the chance of rare events in the output. In the context of MTPP, the next event prediction outputs a single mark and a single time, named NEP in our paper. In the output of NEP, the events with much fewer samples are dominated by events with more samples, i.e., *rare mark missing issue*. Following rare event forecasting, undersampling or oversampling should be able to mitigate the issue in NEP. To evaluate the effectiveness, we have conducted additional experiments where the MTPP model is SAHP [2], a widely accepted baseline in MTPP research.
>
> For undersampling, we reduce the frequency of other marks to ensure they have the same number of training events as the most rare mark. For oversampling, we increase the frequency of other marks so that they have the same number of training events as the most frequent mark. The performances reported in Tables A and B are for oversampling and reported in Tables C and D for undersampling. Oversampling is more impactful to the accuracy of rare marks than undersampling so the following discussion focuses on oversampling. For time prediction, oversampling cannot reliably improve the accuracy for rare marks (compare SAHP in Table A below and Table 1 in our paper). For mark prediction, oversampling can improve the accuracy for rare marks by sacrificing the accuracy for frequent marks (compare SAHP in Table B below and Table 3 in our paper).
>
> The results verify our belief that techniques like undersampling or oversampling cannot remove the root cause of the rare mark missing issue in NEP. Therefore, we target RM-NEP. Specifically, RM-NEP outputs $p^*(m)$ (the probability of the next event being $m$) and $\bar{t}_m$ (time the next event will happen provided its mark is $m$) for every mark $m$. Since every mark is already in the result of RM-NEP, we focus on increasing the accuracy of $p^*(m)$ and $\bar{t}_m$ rather than increasing the chance of rare events in the next event prediction. To this end, we proposed IFNMTPP. Since RM-NEP returns $p^*(m)$ and $\bar{t}_m$ for every mark, it is straightforward to figure out which mark has the highest $p^*(m)$ and its time as the single mark and the single time like the output of NEP. We compared such a single mark and the single time to evaluate the accurate $p^*(m)$ and $\bar{t}_m$ for every mark. The experiments in our paper already demonstrated the advantages of our method.
>
> For mark prediction, SAHP with oversampling achieves a higher accuracy for rare marks than our IFNMTPP only by significantly sacrificing the accuracy for frequent marks (compare SAHP in Table B below and IFNMTPP in Table 3 in our paper). For time prediction, compared to SAHP with oversampling, our IFNMTPP has a significant advantage on all datasets for both rare and frequent marks (compare SAHP in Table A below and IFNMTPP in Table 1 in our paper).
>
> [1] Chathurangi Shyalika, Ruwan Wickramarachchi, and Amit P. Sheth. 2024. A Comprehensive Survey on Rare Event Prediction. ACM Comput. Surv. 57, 3, Article 70 (March 2025), 39 pages. https://doi.org/10.1145/3699955
>
> [2] Zhang, Q., Lipani, A., Kirnap, O., and Yilmaz, E. Self-Attentive Hawkes Process. In Proceedings of the 37th International Conference on Machine Learning, pp. 11183–11193. PMLR, November 1. ISSN: 2640-3498.
>
> Table A: Time prediction performance of SAHP with oversampling on real-world datasets measured by MMAE, lower is better.
> |                         | BO     | Retweet | SO     | Taobao | USearthquake | Yelp   |
> |-------------------------|--------|---------|--------|--------|--------------|--------|
> | $MMAE_{\mathrm{M}}$     | 4.0410 | 3842.4  | 0.8040 | 1.3655 | 0.7700       | 5.3254 |
> | $MMAE_{\mathrm{M}_{r}}$ | 2.8396 | 3594.1  | 0.7885 | 1.4450 | 0.7940       | 5.3744 |
> | $MMAE_{\mathrm{M}_{f}}$ | 5.7506 | 3973.0  | 0.8590 | 0.5522 | 0.7392       | 5.2286 |
>
>
> Table B: Mark prediction performance of SAHP with oversampling on real-world datasets measured by macro-F1, higher is better.
> |                | BO     | Retweet | SO     | Taobao | USearthquake | Yelp   |
> |----------------|--------|---------|--------|--------|--------------|--------|
> | All Marks      | 0.6007 | 0.3071  | 0.0847 | 0.1520 | 0.0978       | 0.2397 |
> | Rare Marks     | 0.7471 | 0.1783  | 0.1223 | 0.1148 | 0.0846       | 0.3101 |
> | Frequent Marks | 0.7341 | 0.2853  | 0.0537 | 0.1030 | 0.0974       | 0.0956 |

---

> ### Author Response · Authors · 2024-11-23
> **Rebuttal to Reviewer i8f9 (2/4)**
>
> Table C: Time prediction performance of SAHP with undersampling on real-world datasets measured by MMAE, lower is better.
> |                         | BO     | Retweet | SO     | Taobao | USearthquake | Yelp   |
> |-------------------------|--------|---------|--------|--------|--------------|--------|
> | $MMAE_{\mathrm{M}}$     | 4.2071 | 3493.3  | 0.9230 | 0.6536 | 0.8571       | 5.3703 |
> | $MMAE_{\mathrm{M}_{r}}$ | 3.2666 | 3619.7  | 0.9690 | 0.6759 | 0.8629       | 5.4125 |
> | $MMAE_{\mathrm{M}_{f}}$ | 5.4184 | 3431.8  | 0.7826 | 0.3825 | 0.8495       | 5.2868 |
>
> Table D: Mark prediction performance of SAHP with undersampling on real-world datasets measured by macro-F1, higher is better.
> |                | BO     | Retweet | SO     | Taobao | USearthquake | Yelp   |
> |----------------|--------|---------|--------|--------|--------------|--------|
> | All Marks      | 0.5987 | 0.2932  | 0.0414 | 0.0680 | 0.1186       | 0.2566 |
> | Rare Marks     | 0.7316 | 0.2213  | 0.0842 | 0.0466 | 0.0857       | 0.2571 |
> | Frequent Marks | 0.7443 | 0.2684  | 0.0187 | 0.0596 | 0.1192       | 0.1678 |
>
> > "W2: The motivation of RM-NEP is not convincing. (i) If a mark is rare (i.e., it occurs very few times in the history). Then, it can be dominated by frequent marks in the prediction. This phenomenon is completely normal. (ii) If a mark is rare and important compared to other marks, why don’t we only consider that mark as a single variable so that there is no imbalance anymore?"
>
>
> For (i), we agree. The frequent marks dominate the rare marks in the result of the next event prediction. This phenomenon is normal but undesired. It leads to a situation where the prediction is a frequent mark but a rare event happens. If the rare event is critical, the consequence of missing it in prediction is risky as explained in our paper (section "Introduction").
>
> If our understanding is correct, "considering that rare mark as a single variable" means filtering out events of other marks from the event sequence and then training a prediction model on the events of that rare mark. This method is irrational even though one can do it. According to the definition of MTPP, all events that happened previously are assumed correlated to the following events. If filtering out the events of other marks, it implies most events are deleted and the remaining events of that rare mark are limited. Training a prediction model based on limited events means significant information is discarded without scrutiny and thus irrational.
>
> > "W3: The paper is not self-contained. For example, how the existing studies solve NEP is not clear. The authors only list a large number of papers in the Related Work section. Similarly, how the existing studies model MTPP is not clear. The authors only list a large number of papers in the Introduction section. A summarization and comparison are needed to provide a better understanding."
>
> We reckon that the presentation order in our paper causes the reviewer's concern. In the Related Work section, we summarize existing works and their strategies for MTPP modeling. After that, the Preliminary section introduces the details of MTPP modeling and the three methods used by existing studies to solve NEP using MTPP models. To improve the paper's readability, we have placed the Preliminary section before the Related Work section in the revised version.
>
> > "W4: Some words are hard to understand. For example, RMTPP is not defined."
>
> RMTPP [3] is an MTPP modeling approach. It is briefly introduced in the Related Work section. Also, we explained why RMTPP is not used as a baseline in the "Baseline Models" part of the Experiments section. Following the comments, we have added more details of RMTPP in the Related Work section of the revised version.
>
> [3] Du, N., Dai, H., Trivedi, R., Upadhyay, U., Gomez-Rodriguez, M., and Song, L. Recurrent Marked Temporal Point Processes: Embedding Event History to Vector. In Proceedings of the 22nd ACM SIGKDD International Conference on Knowledge Discovery and Data Mining, pp. 1555–1564, New York, New York USA, 2016. ACM. doi: 10.1145/2939672.2939875.

---

> ### Author Response · Authors · 2024-11-23
> **Rebuttal to Reviewer i8f9 (3/4)**
>
> > "W5: Some notations are not defined. For example, what is $\tau$?"
>
> The integral of a real-valued function $f(x)$ with respect to a real variable $x$ on an interval $[a, b]$ is written as
> $$
>     \int_{a}^{b}{f(x)dx}
> $$
> The function $f(x)$ is called the integrand, the points $a$ and $b$ are called the limits (or bounds) of integration, and the integral is said to be over the interval $[a, b]$, called the interval of integration. Using Equation (3) in our paper as an example:
> $$
>     p^*(m) = \int_{t_l}^{+\infty}{p^*(m, \tau)d\tau}
> $$
> $\tau$ is a variable of the integrand $p^*(m, \tau)$, meaning time. $t_l$ indicates the lower bound of the integration.
>
> After checking our paper against the comment, we identified and fixed a typo in Equation (2), where the notation of the intensity function in the integral should be $\lambda^*(n, \tau)$, not $\lambda^*(n, t)$. We guess this typo is the reason behind this comment. Moreover, we have explained $\tau$ after Equation (2) in the revised version.
>
>
> > "W6: Intuitively, can we solve the problem by undersampling dominating marks?"
>
> Yes, undersampling the dominating mark can mitigate the rare mark missing issue in NEP. Please see more discussions in our response to the first comment.
>
>
> > "W7: I cannot understand lines 297-299. If $t=t_l$ then the integration equals 0."
>
> Please note line 297-299 (in the paragraph under Equation (8)), we explain $\Gamma^*(m, t)$ in Equation (8) and its relationship with Equation (3). Specifically, $\Gamma^*(m, t)$ is the integration starting from time $t$, any time after $t_l$ or $t_l$, to positive infinity:
> $$
>     \Gamma^*(m, t) = \int_{t}^{+\infty}{p^*(m, \tau)d\tau}
> $$
>   Equation (3) is:
> $$
>     p^*(m) = \int_{t_l}^{+\infty}{p^*(m, \tau)d\tau}
> $$
> Clearly, $p^*(m)$ is equivalent to $\Gamma^*(m, t_l)$. If we can solve $\Gamma^*(m, t)$, we can solve Equation (3) by simply setting $t=t_l$. To make it clearer, we have polished the paragraph under Equation (8) as below:
>
> "For each mark $m\in \mathrm{M}$, $\Gamma^*(m, t)$ is the integration starting from time $t$, any time after $t_l$ or $t_l$, to positive infinity. $\Gamma^*(m, t)$ is monotonically decreasing as its derivative $-p^*(m, t)$ is always smaller than 0. By definition, $p^*(m)$ in Equation (3) is equivalent to $\Gamma^*(m, t_l)$. That is, if we can solve $\Gamma^*(m, t)$, $p^*(m)$ can be solved by setting $t=t_l$. It means two different target integrals in Equation (3) and Equation (4) are now unified into one, i.e., $\Gamma^*(m, t)$."
>
>
> > "W8: The main idea of using integral-free comes from FullyNN by using IEM. Basically, the authors adapt it to marked events, which is straightforward. "
>
> As introduced in section "Integral-Free Neural Marked Temporal Point Process (IFNMTPP)", IEM consists of multiple fully-connected layers with non-negative weights and monotonic-increasing activation functions. IEM cannot achieve the integral-free solution by itself. Instead, the integral-free solution is achieved based on all components of IFNMTPP as a whole.
>
> Even though structurally similar, extending FullyNN to IFNMTPP is not straightforward. IFNMTPP and FullyNN solve different integration functions. FullyNN aims to solve $\lambda^*(t)$ by estimating its integral $\Lambda^*(t)$ where events have the identical mark. IFNMTPP aims to solve $p^*(m, t)$ by estimating its integral $\Gamma^*(m, t)$ where events have different marks. $\lambda^*(t)$ and $p^*(m, t)$ are different concepts and their relationship is defined by Equation (2). Extending FullyNN to IFNMTPP needs to address these differences properly. Various methods have been investigated including FENN, one of the baselines, before IFNMTPP.
>
> Compared with how IFNMTPP works, it is more important why IFNMTPP is necessary as a part of the holistic solution for RM-NEP, which involves the improper integration of two different functions in Equation (3) and Equation (4), respectively. Separately solving each integration problem is computationally inefficient. To address the challenge, we transform the improper integration of two different functions into one, namely $\Gamma^*(m, t)$, for an effective solution. To solve $\Gamma^*(m, t)$, IFNMTPP is designed deliberately.
>
>
> > "W9: The authors do not prove why using IEM can achieve the integral-free solution."
>
> As introduced in the section "Integral-Free Neural Marked Temporal Point Process (IFNMTPP)", IEM consists of multiple fully-connected layers with non-negative weights and monotonic-increasing activation functions. IEM cannot achieve the integral-free solution by itself. Instead, the integral-free solution is achieved based on all components of INFMTPP as a whole to approximate $\Gamma^*(m, t)$. To clarify how IFNMTPP approximates $\Gamma^*(m, t)$ in the revised version, we have updated Figure 2 by including more structural details of IEM, and have added more explanations in the paragraph before Equation (9).

---

> ### Author Response · Authors · 2024-11-23
> **Rebuttal to Reviewer i8f9 (4/4)**
>
> > "W10: There is no ablation study. For example, what is the performance of TFNMTPP with different imbance ratios?"
>
> For a comprehensive evaluation of IFNMTPP across datasets of varying levels of imbalance, we purposedly include datasets like BookOrder with very low levels of imbalance and datasets like StackOverflow with very high levels of imbalance to show that our approach is robust to datasets with different imbalance ratios.

---

> ### Author Response · Authors · 2024-11-27
> **Additional experiment results (1/2)**
>
> To further answer the question "Intuitively, can we solve the problem by undersampling dominating marks?", we conducted additional experiments using undersampling and oversampling on all baseline MTPP models.
>
> For undersampling in each dataset, we reduce the frequency of other marks to ensure they have the same number of training events as the most rare mark. For oversampling in each dataset, we increase the frequency of other marks so that they have the same number of training events as the most frequent mark. The performances reported in Tables E and F are for oversampling and reported in Tables G and H for undersampling. Oversampling is more impactful to the accuracy of rare marks than undersampling so the following discussion focuses on oversampling.
>
> For time prediction, oversampling improves the accuracy for rare marks on some datasets but impairs the accuracy on other datasets (compare the corresponding baselines in Table E below and Table 1 in our paper). For mark prediction, oversampling cannot improve the accuracy for rare marks by sacrificing the accuracy for frequent marks (compare the corresponding baselines in Table F below and Table 3 in our paper). Compared with datasets with a high imbalance level, undersampling or oversampling generally has a limited impact on the mark and time prediction accuracy for the datasets with a low imbalance level like BookOrder.
>
> The results verify our belief that techniques like undersampling or oversampling cannot remove the root cause of the rare mark missing issue in NEP. It may be possible to improve the performance of undersampling or oversampling by using different sampling rates. However, such a hyperparameter tuning is challenging. Therefore, we targeted RM-NEP and proposed a new MTPP model, i.e., IFNMTPP. Compared with baselines with oversampling, the performance of our IFNMTPP remains its superiority in time prediction and advantage in mark prediction.
>
> Table E: Time prediction performance of baselines with oversampling on real-world datasets measured by MMAE, lower is better.
>
> ||BO|Retweet|SO|Taobao|USearthquake|Yelp|
> |-|-|-|-|-|-|-|
> |FENN|$MMAE_{\mathrm{M}}$|124.35|4430.3|1.1931|3.1415|6.5182|
> ||$MMAE_{\mathrm{M}_{r}}$|124.02|7312.9|1.4128|3.1065|6.5512|
> ||$MMAE_{\mathrm{M}_{f}}$|124.68|3448.3|0.6715|3.7574|6.4527|
> |FullyNN|$MMAE_{\mathrm{M}}$|125.74|4745.3|0.7320|4.6986|6.8449|
> ||$MMAE_{\mathrm{M}_{r}}$|124.02|7190.6|0.7561|4.6690|6.9323|
> ||$MMAE_{\mathrm{M}_{f}}$|124.68|3930.7|0.6556|5.1969|6.6734|
> |SAHP|$MMAE_{\mathrm{M}}$|4.0410|3842.4|0.8040|1.3655|5.3254|
> ||$MMAE_{\mathrm{M}_{r}}$|2.8396|3594.1|0.7885|1.3310|5.2286|
> ||$MMAE_{\mathrm{M}_{f}}$|1.5537|3567.2|0.7018|3.7999|5.2680|
> |THP|$MMAE_{\mathrm{M}}$|1.4510|3819.9|0.6986|2.5922|5.3533|
> ||$MMAE_{\mathrm{M}_{r}}$|1.3551|4380.3|0.6880|2.5310|5.3968|
> ||$MMAE_{\mathrm{M}_{f}}$|1.5537|3567.2|0.7018|3.7999|5.2680|
> |Marked-LNM|$MMAE_{\mathrm{M}}$|1.2273|6667.9|10.401|3.8806|5.3660|
> ||$MMAE_{\mathrm{M}_{r}}$|1.1429|57940|9.7496|3.2644|5.4198|
> ||$MMAE_{\mathrm{M}_{f}}$|1.3180|2262|12.958|61.718|5.2601|
>
>
> Table F: Mark prediction performance of baselines with oversampling on real-world datasets measured by macro-F1, higher is better.
> |Model||BO|Retweet|SO|Taobao|USearthquake|Yelp|
> |-|-|-|-|-|-|-|-|
> |FENN|All Marks|0.3595|0.1882|0.0092|0.0051|0.0979|0.1566|
> ||Rare Marks|0.0273|0.4261|0.0471|0.0166|0.0000|0.2942|
> ||Frequent Marks|0.5502|0.1665|0.0071|0.0000|0.1419|0.0132|
> |FullyNN|All Marks|0.3339|0.2316|0.0121|0.0194|0.1621|0.0953|
> ||Rare Marks|0.0000|0.0000|0.0000|0.0437|0.0000|0.2634|
> ||Frequent Marks|1.0000|0.3282|0.0287|0.0000|0.2316|0.0000|
> |SAHP|All Marks|0.6007|0.3071|0.0847|0.1520|0.0978|0.2397|
> ||Rare Marks|0.7471|0.1783|0.1223|0.1148|0.0846|0.3101|
> ||Frequent Marks|0.7341|0.2853|0.0537|0.1030|0.0974|0.0956|
> |THP|All Marks|0.5857|0.0274|0.0816|0.0140|0.0791|0.1606|
> ||Rare Marks|0.7114|1.0000|0.2032|0.0261|0.0019|0.3983|
> ||Frequent Marks|0.7419|0.0000|0.0410|0.0000|0.1090|0.0000|
> |Marked-LNM|All Marks|0.6036|0.1995|0.0805|0.2817|0.0930|0.2616|
> ||Rare Marks|0.7451|0.1768|0.2655|0.1932|0.0825|0.1619|
> ||Frequent Marks|0.7551|0.3966|0.0304|0.1819|0.0863|0.3737|

---

> ### Author Response · Authors · 2024-11-27
> **Additional experiment results (2/2)**
>
> Table G: Time prediction performance of baselines with undersampling on real-world datasets measured by MMAE, lower is better.
> |||BO|Retweet|SO|Taobao|USearthquake|Yelp|
> |--------|-----|-------------|------------|------------|-------------|--------------|------------|
> |FENN|$MMAE_{\mathrm{M}}$|124.28|4555.4|0.9906|3.0623|0.8425|6.2257|
> ||$MMAE_{\mathrm{M}_{r}}$|123.98|6686.3|1.1165|3.0265|0.8388|6.2551|
> ||$MMAE_{\mathrm{M}_{f}}$|124.28|3768.1|0.7075|3.6969|0.8492|6.1674|
> |FullyNN|$MMAE_{\mathrm{M}}$|125.19|4681.5|0.7976|3.1756|0.8208|6.5248|
> ||$MMAE_{\mathrm{M}_{r}}$|125.11|6831.0|0.8289|3.1435|0.8263|6.5552|
> ||$MMAE_{\mathrm{M}_{f}}$|125.26|3903.4|0.7002|3.7350|0.8137|6.4644|
> |SAHP|$MMAE_{\mathrm{M}}$|4.2071|3493.3|0.9230|0.6562|0.8495|5.2868|
> ||$MMAE_{\mathrm{M}_{r}}$|1.2763|4054.3|0.7047|3.1064|0.9611|5.4198|
> ||$MMAE_{\mathrm{M}_{f}}$|1.4518|3074.6|0.7321|3.8405|0.9022|5.2919|
> |THP|$MMAE_{\mathrm{M}}$|1.3612|3371.6|0.7108|3.1454|0.9354|5.3769|
> ||$MMAE_{\mathrm{M}_{r}}$|1.2763|4054.3|0.7047|3.1064|0.9611|5.4198|
> ||$MMAE_{\mathrm{M}_{f}}$|1.4518|3074.6|0.7321|3.8405|0.9022|5.2919|
> |Marked-LNM|$MMAE_{\mathrm{M}}$|1.2091|17799|4.6132|18.851|579.72|5.3999|
> ||$MMAE_{\mathrm{M}_{r}}$|1.1243|22325|3.9546|17.287|656.09|5.4608|
> ||$MMAE_{\mathrm{M}_{f}}$|1.3003|15892|7.7888|75.306|491.54|5.2800|
>
>
> Table H: Mark prediction performance of baselines with undersampling on real-world datasets measured by macro-F1, higher is better.
> |   |  | BO | Retweet | SO | Taobao | USearthquake | Yelp |
> | --- | --- | --- | --- | --- | --- | --- | --- |
> | FENN | All Marks | 0.3623 | 0.1224 | 0.0736 | 0.1395 | 0.0733 | 0.2382 |
> |  | Rare Marks | 0.0335 | 0.6670 | 0.0000 | 0.0000 | 0.4199 | 0.3533 |
> |  | Frequent Marks | 0.5008 | 0.0965 | 0.1500 | 1.0000 | 0.0547 | 0.0337 |
> | FullyNN | All Marks | 0.3339 | 0.2316 | 0.0121 | 0.0194 | 0.1621 | 0.0953 |
> |  | Rare Marks | 0.0000 | 0.0000 | 0.0000 | 0.0437 | 0.0000 | 0.2634 |
> |  | Frequent Marks | 1.0000 | 0.3282 | 0.0287 | 0.0000 | 0.2316 | 0.0000 |
> | SAHP | All Marks | 0.5987 | 0.2932 | 0.0414 | 0.0680 | 0.1186 | 0.2566 |
> |  | Rare Marks | 0.7316 | 0.2213 | 0.0842 | 0.0466 | 0.0857 | 0.2571 |
> |  | Frequent Marks | 0.7443 | 0.2684 | 0.0187 | 0.0596 | 0.1192 | 0.1678 |
> | THP | All Marks | 0.5914 | 0.2185 | 0.0307 | 0.0150 | 0.1054 | 0.2344 |
> |  | Rare Marks | 0.7974 | 0.0801 | 0.1972 | 0.0228 | 0.0667 | 0.3137 |
> |  | Frequent Marks | 0.6709 | 0.2574 | 0.0000 | 0.0029 | 0.1138 | 0.0833 |
> | Marked-LNM | All Marks | 0.6008 | 0.1988 | 0.0559 | 0.2292 | 0.1255 | 0.2494 |
> |  | Rare Marks | 0.7235 | 0.2540 | 0.1174 | 0.1943 | 0.0656 | 0.1697 |
> |  | Frequent Marks | 0.7714 | 0.1737 | 0.0214 | 0.1227 | 0.1405 | 0.3493 |

---

### Comment · Area_Chair_BKT8 · 2024-11-25
**Acknowledge the author responses**

Dear Reviewers,

Thank you very much for your effort. As the discussion period is coming to an end, please acknowledge the author responses and adjust the rating if necessary.

Sincerely,
AC

---

### Author Response · Authors · 2024-11-27
**Overall response**

We appreciate all four thorough reviews of our paper. Based on these reviews, we made the following changes to our paper (all line numbers and section numbers correspond to the revised version):

- As a response to the comment of reviewer i8f9, o69X, and oDxB. We updated Figure 2 in section 3.2 "Integral-Free Neural Marked Temporal Point Process (IFNMTPP)" by including more structural details of IFNMTPP.
- To address the reviewer's concern about the marginal performance improvement of our method in mark prediction, we rewrite section 4.2 to provide more analyses of the results in Table 3.
- As a response to weakness 3 of reviewer i8f9, we replaced citations in the Introduction section with a review paper by Shchur et al. We also changed the Preliminary section from section 3 to section 2 and placed the Related Work section after the Experiment section for better readability.
- As a response to weakness 4 of reviewer i8f9, we add details for RMTPP (Recurrent Marked Temporal Point Process). Specifically, we added the full name of RMTPP when we first mentioned it in line 339 and introduced it in the Related Work section, starting from line 496.
- As a response to weakness 5 of reviewer i8f9, we explain the meaning of $\tau$ at line 118 by adding "where $\tau$ means time."
- As a response to weakness 7 of reviewer i8f9, we polished the paragraph under Equation (8) from line 257 to clarify that we are discussing $\Gamma^*(m, t)$, not $F^*(m, t)$.
- As a response to weakness 2 of reviewer ZLCK, we explain why $\Gamma^*(m, t)$ is a monotonically decreasing function at line 258 by adding "$\Gamma^*(m, t)$ is monotonically decreasing as its derivative $-p^*(m, t)$ is always smaller than 0." We also modified the paragraph starting from line 288 to explain how IFNMTPP approximates $\Gamma^*(m, t)$ and why a monotonically decreasing function is appropriate for IFNMTPP.
- As a response to weakness 3 of reviewer o69X, we have removed the comma after the question mark in "...what is the probability that the mark of the next event is $m$? and if $m$, when will the next event happen?..." at line 20, line 78, line 195, and line 535.

---

### Comment · Area_Chair_BKT8 · 2024-11-28
**Discussion needed**

Dear Reviewers,

As you are aware, the discussion period has been extended until December 2. Therefore, I strongly urge you to participate in the discussion as soon as possible if you have not yet had the opportunity to read the authors' response and engage in a discussion with them. Thank you very much.

Sincerely,
Area Chair

---

### Meta-Review · Area_Chair_BKT8 · 2024-12-19

**Metareview:**

This paper presents a rare event forecasting problem, RM-NEP, in the context of the marked temporal point process (MTPP). The reviewers agreed that the paper is well-written and the problem is very interesting.  However, they also raised several concerns.  Most commonly, the performance improvement is not very impressive.  Regarding the novelty, the reviewers pointed out that RM-NEP is not significantly different from the MTPP.  Although the authors did not agree with this point, I think that the reviewers made a reasonable point.  This paper is indeed a borderline paper.  However, in my batch, there are quite sufficient papers which are strongly supported by the reviewers.  Thus, I would like to recommend a reject.  If there is room, the senior AC can change my recommendation.

**Additional Comments On Reviewer Discussion:**

* Reviewer i8f9 acknowledged the authors' responses, but he/she was not fully satisfied with the authors' responses (regarding novelty and evaluation results).  This reviewer's opinion makes sense.

* Reviewer ZLCK would like to support the acceptance of this paper although the performance improvement is marginal.

* Reviewer oDxB was satisfied with the authors' responses.

---

### Decision · Program_Chairs · 2025-01-22

Reject